Journal of Data-centric Machine Learning Research (2026)        Submitted 1/25; Revised 10/25; Published 2/26

# ARLBench: Flexible and Efficient Benchmarking for Hyperparameter Optimization in Reinforcement Learning

**Jannis Becktepe**[*1,2,†]                          JANNIS.BECKTEPE@TU-DORTMUND.DE

**Julian Dierkes**[*3]                               DIERKES@AIM.RWTH-AACHEN.DE

**Carolin Benjamins**[4]                             C.BENJAMINS@AI.UNI-HANNOVER.DE

**Aditya Mohan**[4]                                  A.MOHAN@AI.UNI-HANNOVER.DE

**David Salinas**[5]                                 SALINASD@CS.UNI-FREIBURG.DE

**Raghu Rajan**[5]                                   RAJANR@CS.UNI-FREIBURG.DE

**Frank Hutter**[5,6]                                FH@CS.UNI-FREIBURG.DE

**Holger H. Hoos**[3]                                HH@AIM.RWTH-AACHEN.DE

**Marius Lindauer**[4,7]                             M.LINDAUER@AI.UNI-HANNOVER.DE

**Theresa Eimer**[4,7]                               T.EIMER@AI.UNI-HANNOVER.DE

[1] *TU Dortmund University,* [2] *Lamarr Institute for Machine Learning and Artificial Intelligence,*
[3] *RWTH Aachen University,* [4] *Leibniz University Hannover,* [5] *University of Freiburg,*
[6] *ELLIS Institute Tübingen,* [7] *L3S Research Center*
[*]Both authors contributed equally to this work. [†] Work done at Leibniz University Hannover.

**Reviewed on OpenReview:** `https://openreview.net/forum?id=i6iafxkfFF`

**Editor:** Yang Liu

## Abstract

Hyperparameters are a critical factor in reliably training well-performing reinforcement learning (RL) agents. Unfortunately, developing and evaluating automated approaches for tuning such hyperparameters is both costly and time-consuming. As a result, such approaches are often only evaluated on a single domain or algorithm, making comparisons difficult and limiting insights into their generalizability. We propose ARLBench, a benchmark for hyperparameter optimization (HPO) in RL that allows comparisons of diverse HPO approaches while being highly efficient in evaluation. To enable research into HPO in RL, even in settings with low compute resources, we select a representative subset of HPO tasks spanning a variety of algorithm and environment combinations. This selection allows for generating a performance profile of an automated RL (AutoRL) method using only a fraction of the compute previously necessary, enabling a broader range of researchers to work on HPO in RL. With the extensive and large-scale dataset on hyperparameter landscapes that our selection is based on, ARLBench is an efficient, flexible, and future-oriented foundation for research on AutoRL. Both the benchmark and the dataset are available at `https://github.com/automl/arlbench`.

**Keywords:** Automated Reinforcement Learning, Reinforcement Learning, Hyperparameter Optimization, Automated Machine Learning, Benchmarking

## 1 Introduction

Deep reinforcement learning (RL) algorithms require careful configuration of many different design decisions and hyperparameters to reliably work in practice (Farsang and Szegletes, 2021; Pislar et al., 2022), such as learning rates (Gulde et al., 2020) or batch sizes (Obando-Ceron et al., 2023). Automated RL (AutoRL; Parker-Holder et al. (2022)), a sub-field of automated machine learning (AutoML), makes these design decisions in a data-driven manner. In fact, recent work has shown that such a data-driven approach offers the best way of navigating hyperparameters in RL (Zhang et al., 2021; Eimer et al., 2023), due to the complex and changing hyperparameter optimization landscapes encountered (Mohan et al., 2023).

Research on hyperparameter optimization (HPO) for RL has been gaining traction in recent years (Jaderberg et al., 2017; Parker-Holder et al., 2020; Franke et al., 2021; Wan et al., 2022). While such approaches promise to streamline the application of RL by providing users with well-performing hyperparameter configurations for their RL tasks, it is hard to discern their actual quality; each HPO method is usually evaluated on a limited number of environments, combined with a different HPO configuration space (see, e.g., the differences between Parker-Holder et al. (2020) and Shala et al. (2024)). This inability to compare HPO approaches and AutoRL approaches more broadly leads to a lack of clarity and, ultimately, a lack of adoption of an approach that shows great promise in making RL overall more efficient and easier to apply.

One reason for the inconsistent evaluations in the current HPO literature is the wealth of RL algorithms and environments, each with its own challenges. While some environments require the processing of image observations (Bellemare et al., 2013; Cobbe et al., 2020), others focus more on finding the optimal solutions in settings with sparse reward signals (Nikulin et al., 2023). It is fundamentally unclear which environment and algorithm combinations should be considered representative tasks for the current scope of RL research and thus useful as evaluation settings for AutoRL approaches.

We focus on the following question: *Which environments should we evaluate a given RL algorithm on to obtain a reliable performance estimate of an AutoRL method?* To answer it, we first implement highly efficient and configurable versions of three popular RL algorithms: DQN (Mnih et al., 2015), PPO (Schulman et al., 2017), and SAC (Haarnoja et al., 2018). We subsequently generate hyperparameter landscapes across a diverse range of commonly used environment domains, specifically Arcade Learning Environment (ALE; Bellemare et al. (2013)) games, Classic Control and Box2D simulations (Brockman et al., 2016; Towers et al., 2023), Brax robot walkers (Freeman et al., 2021), and XLand-Minigrid exploration tasks (Nikulin et al., 2023) to conduct a large-scale analysis. This study, which we publish as a meta-dataset, allows us to assess the performance of given hyperparameter configurations for each algorithm and environment.

Based on the scores in the generated landscapes, we follow the method proposed by Aitchison et al. (2023) to find the subset of environments with the highest capability for predicting the average performance across all environments in order to model the RL task space. This subset thus matches the tasks the RL community cares about better than previous work on HPO for RL, while reducing computational demands for evaluation. This provides the research community with an empirically sound benchmark for HPO, which

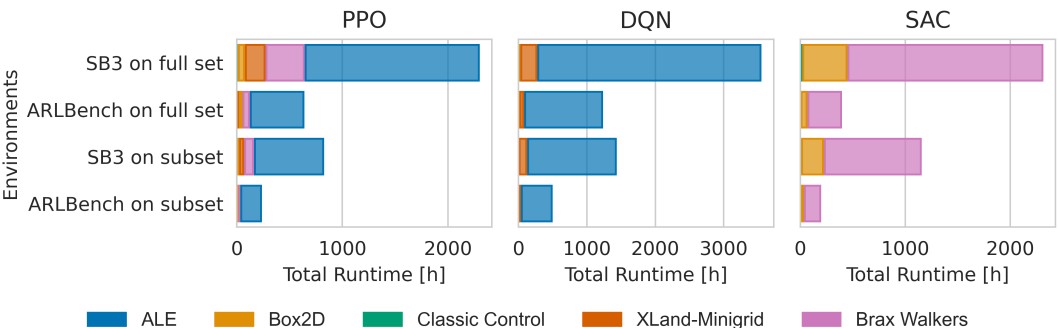

Figure 1: Running time comparison for an HPO method of 32 RL runs using 10 seeds each on the full environment set and our subsets between ARLBench and StableBaselines3 (SB3; Raffin et al. (2021)). This results in speedup factors due to JAX of 3.59 for PPO, 2.87 for DQN, and 5.78 for SAC of ARLBench, compared to SB3 on the full set. The subset selection further decreases the running time by a factor of 2.67 for PPO, 2.49 for DQN, and 2.0 for SAC. Comparing ARLBench on the subset to SB3 on the full set, the total speedups are 9.6 for PPO, 7.14 for DQN, and 11.61 for SAC. Running time comparisons for each environment category can be found in Appendix E. Note that the bars for some domains, especially on ARLBench, may be very small due to low running time.

we dub **ARLBench**. It is highly efficient, taking only 937 GPU hours to evaluate an HPO budget of 32 full RL trainings using 10 seeds each on all three algorithm subsets. StableBaselines (SB3; Raffin et al. (2021)) on the full set of environments would take 8 163 GPU hours, resulting in average speedup factors of 9.6 for PPO, 7.14 for DQN, and 11.61 for SAC as shown in Figure 1.

ARLBench is designed with current AutoRL and AutoML methods in mind; partial execution as used in many contemporary HPO methods (Li et al., 2017; Awad et al., 2021; Lindauer et al., 2022) is built into the benchmark structure just like dynamic optimization in arbitrary intervals, as, e.g., in population-based training (PBT; Jaderberg et al. (2017)) variations. Moreover, various training data from ARLBench, including performance measures such as evaluation rewards and gradient history, can be used in adaptive HPO methods. ARLBench additionally supports large configuration spaces, making most low-level design decisions and architectures configurable for each algorithm. This flexibility and running time efficiency spawns a range of new insights into approaches to AutoRL.

In short, our key contributions are: (i) A highly efficient benchmark for HPO in RL, which natively supports diverse categories of HPO approaches; (ii) an environment subset selection for standardized comparisons that covers the RL task space, both (i) and (ii) together improving computational feasibility by an order of magnitude; (iii) a set of performance data on our benchmark with over 100 000 total runs spanning various RL algorithms, environments, seeds, and configurations (equivalent to 32 588 GPU hours).

## 2 Related Work: Benchmarking HPO for RL

Several works study the impact that such hyperparameter settings have on RL algorithms (Henderson et al., 2018; Andrychowicz et al., 2021; Obando-Ceron and Castro, 2021; Obando-Ceron et al., 2023) and show that they mostly do not transfer across environments (Ceron et al., 2024; Patterson et al., 2024). Further, it has been suggested that several algorithmic performance improvements may be a result of an increased reliance on hyperparameter tuning (Adkins et al., 2024). Automated configuration of these algorithms, on the other hand, is not as common, especially compared to the body of work in AutoML (Hutter et al., 2019). Previous work has shown, however, that HPO approaches can find high-performing hyperparameter configurations quite efficiently (Xu et al., 2018; Parker-Holder et al., 2020; Zhang et al., 2021; Franke et al., 2021; Flennerhag et al., 2022; Wan et al., 2022). These approaches range from standard HPO, including multi-fidelity optimization (Falkner et al., 2018; Awad et al., 2021), and algorithm configuration tools from AutoML (Schede et al., 2022; Dierkes et al., 2024) to novel strategies aiming to adapt to the dynamic nature of RL algorithms. Most popular is the PBT line of work (Jaderberg et al., 2017; Wan et al., 2022; Coward et al., 2024), which evolves hyperparameter schedules via a population of agents, resulting in a dynamic configuration strategy. For adaptive dynamic HPO, second-order optimization can be used to learn hyperparameter schedules online (Xu et al., 2018; Zahavy et al., 2020; Flennerhag et al., 2022). Further, gradient-free methods have been explored (Vincent et al., 2024; Paul et al., 2019). Most of these, however, are not directly comparable due to different algorithms, environments, and configuration spaces in their experiments, making it difficult to find clear state-of-the-art and thus promising directions for future work (Eimer et al., 2023).

Besides this lack of comparisons in HPO for RL, the cost of training and evaluation is a significant factor hindering progress in the field. Tabular benchmarks (Ying et al., 2019; Klein and Hutter, 2019) offer a low-cost option when benchmarking HPO. Such benchmarks are essentially databases, from which the results of running a given algorithm are looked up rather than performing actual runs. Currently, the only benchmark library for HPO in RL is a tabular benchmark: HPO-RL-Bench (Shala et al., 2024). It contains results for five RL algorithms on 22 different environments with three random seeds each. HPO-RL-Bench offers significantly reduced configuration spaces of only up to three hyperparameters, narrowed down from typically larger spaces of 10 to 13 hyperparameters, e.g., in ARLBench and SB3 (Raffin et al., 2021), and is based solely on a pre-computed dataset. Its dynamic option is further reduced to only two hyperparameters, each with three possible values at two switching points. We believe, therefore, that HPO-RL-Bench and ARLBench will fulfill different roles: HPO-RL-Bench can provide zero-cost evaluations of expensive domains, while for ARLBench, we prioritized flexibility in what and when to configure while still allowing fast evaluations.

Benchmarks are essential in the broader AutoML domain; Benchmarks such as HPOBench (Eggensperger et al., 2021), HPO-B (Pineda et al., 2021) and YAHPO-Gym (Pfisterer et al., 2022) have been contributing to research progress in HPO. In contrast, ARLBench focuses exclusively on RL, a domain that has only been included with a single toy scenario in HPOBench so far. Given that Mohan et al. (2023) have shown that the RL HPO landscapes do not seem as benign as Pushak and Hoos (2018) describe the HPO landscapes for supervised

learning overall (see Appendix H.1), it is necessary to offer a dedicated RL benchmark with a diverse task set. The NAS-Bench benchmarks for neural architecture search (NAS) are examples of benchmarking supporting efficient research: NAS-Bench-101 (Ying et al., 2019) is a tabular benchmark, which NAS-Bench-201 (Dong and Yang, 2020) extends to a larger configuration space, and NAS-Bench-301 (Zela et al., 2022) uses this data to propose surrogate models (Eggensperger et al., 2014; Klein et al., 2019) that can predict performance even for unseen architectures. Building on these, several dozens of specialized NAS benchmarks have been developed (Mehta et al., 2022). We expect that benchmarking HPO in RL will similarly become a focal point within the community towards advancing the configuration of RL algorithms.

## 3 Implementing ARLBench

In this section, we discuss the implementation of the ARLBench framework. Notably, we elaborate on essential considerations for the benchmark and its two main components: the AutoRL Environment HPO interface and the RL algorithm implementations.

### 3.1 Benchmark Desiderata for ARLBench

Given the limitations of HPO-RL-Bench (Shala et al., 2024) compared to the kinds of methods we see in HPO for RL, our three main priorities in constructing ARLBench are (i) enabling the large configuration spaces required for RL, (ii) prioritizing fast execution times, and (iii) supporting dynamic and reactive hyperparameter schedules.

**Configuration Space Size.** Eimer et al. (2023) have shown that most hyperparameters contribute to the training success of RL algorithms. Furthermore, our knowledge of how hyperparameters act on RL algorithms continues to expand, most recently, e.g., by showing the importance of batch sizes in certain RL settings (Obando-Ceron et al., 2023). Thus, limiting the configurability of a benchmark will lead to the insights we gather outpacing the benchmarking capabilities of the community. Therefore, we enable large and flexible configuration spaces for all algorithms. To achieve this, however, we cannot simply extend the tabular HPO-RL-Bench, as the computational expense required for larger configuration spaces would grow exponentially in the number of hyperparameters. A long-term solution would be to train surrogate models to predict performance. However, as the data requirements for reliable and dynamic surrogates in RL are presently unclear, we focus on building a good online benchmark first and use it to generate preliminary landscapes. We hope this approach allows the building of better RL-specific surrogate models in future work.

**Running Time.** An alternative to using surrogate models is building an efficient way of evaluating hyperparameter configurations in RL. JAX (Bradbury et al., 2018) enables significant efficiency gains, leading to RL agents training on many domains in mere minutes or seconds (Lu, 2022; Toledo, 2024). We exploit this while providing RL algorithms that are easy to configure for commonly used HPO methods, including multi-objective and multi-fidelity optimization.

**Dynamic Configuration.** Finally, we aim to enable dynamic configurations that allow hyperparameter settings to be adjusted during a single RL training session, recognizing that the optimal hyperparameters can evolve as training progresses (Mohan et al., 2023). One way of doing this is by providing checkpoint capabilities that support the seamless continuation

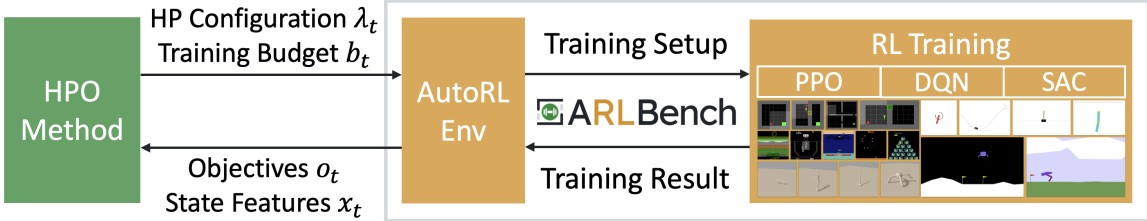

Figure 2: Overview of the ARLBench framework. The AutoRL environment, providing a *Gymnasium*-like interface (Towers et al., 2023), is the interaction point for HPO methods. At optimization step $t$, the optimizer selects a hyperparameter configuration $\lambda_t$ and a training budget (number of steps) $b_t$. Then, the RL algorithm is trained using the given configuration and budget. As a result, the AutoRL environment returns the training result in the form of optimization objectives $o_t$, e.g., the evaluation return and runtime, and state features $x_t$, e.g., gradients during training.

of RL training. Most population-based methods, for example, find schedules with 10 to 20 hyperparameter changes during a single training run (Jaderberg et al., 2017; Parker-Holder et al., 2022; Wan et al., 2022), while other methods, such as hyperparameter adaptation via meta-gradients, can configure much more often and even require information about the current algorithm state.

## 3.2 The HPO Interface: The AutoRL Environment

As shown in Figure 2, the AutoRL Environment is the main building block of ARLBench and connects all the critical parts for HPO in RL. It provides a powerful, flexible, and dynamic interface to support various HPO methods in an interface that, for ease of use, functions similarly to *Gymnasium* (Towers et al., 2023). During the optimization, the HPO method selects a hyperparameter configuration $\lambda_t$ and training budget $b_t$ for the current optimization step $t$. Given these, the AutoRL Environment sets up the algorithm and RL environment and performs the actual RL training. In addition to an evaluation reward, data such as gradients and losses are collected during training. Depending on the user's preferences, the AutoRL Environment then extracts optimization objectives, such as the average evaluation reward, training running time, or carbon emissions (Courty et al., 2024), as well as optional information on the internal state of the RL algorithm, e.g., the variance of the gradients.

The AutoRL Environment supports static and dynamic HPO methods. While static methods start the inner RL training from scratch for each configuration, dynamic approaches can keep the training state, which includes the neural network parameters, optimizer state, and replay buffer. To support the latter, we integrate an easy-to-use yet powerful checkpointing mechanism. This enables HPO methods to restore, duplicate, or checkpoint the training state at any point during the dynamic optimization.

## 3.3 RL Training

To address the computational efficiency of RL algorithms, we implement the entire training pipeline using JAX (Bradbury et al., 2018). We re-implement DQN (Mnih et al., 2015), PPO (Schulman et al., 2017), and SAC (Haarnoja et al., 2018) in order to make them highly configurable, enable dynamic execution, and ensure compatibility with different target environments. Wherever we use code from external sources (Freeman et al. (2021); Lu (2022); Toledo et al. (2023); licensed under Apache-2.0), it is referenced in the code. We compare our implementation to SB3 (Raffin et al., 2021) in Appendix E and find very similar learning curves. We support a range of environment frameworks, particularly *Brax* (Freeman et al., 2021), *Gymnax* (Lange, 2022), *Gymnasium* (Towers et al., 2023), *Envpool* (Weng et al., 2022), and *XLand-Minigrid* (Nikulin et al., 2023). This results in a broad coverage of RL domains, including robotic simulations, grid worlds, and video games, such as the ALE (Bellemare et al., 2013). We ensure compatibility with these different environments and their APIs with our own ARLBench *Environment* class, allowing for future updates and continued support of changing interfaces in RL.

## 4 Finding Representative Benchmarking Settings

Highly efficient implementations are crucial for efficient benchmarking of HPO methods for RL. However, they represent just a fraction of the overall picture: prior work has focused primarily on a single-task domain, due to a lack of insight regarding which RL domains to target. To tackle this issue, we aim to find a subset of RL environments representative of the broader RL field. First, we study the hyperparameter landscapes for a large set of environments using random sampling of configurations. To ensure the feasibility of our experiments in terms of computational resources, we select a representative subset of environments from each domain. In particular, we select a total of 21 environments: five ALE games (Atari-5), three Box2D environments, four Brax walkers, five classic control environments, and four XLand-Minigrid environments (see Appendix D). Then, we use an approach similar to the selection of the Atari-5 (Aitchison et al., 2023) environments, to find a subset of environments for testing HPO approaches in RL. Ultimately, we validate that this subset is representative of the HPO landscape of all RL tasks we consider. For obtaining our results we spend a total of 10 105 h on CPUs and 32 588 h on GPUs (see Appendix I).

### 4.1 Data Collection

For each combination of algorithm and environment, we aim to estimate the hyperparameter landscape, i.e., the relationship between a certain hyperparameter configuration and its performance. Therefore, we run an RL algorithm on 256 Sobol-sampled configurations (Sobol, 1967). With configuration spaces ranging from 10 to 13 hyperparameters, this is roughly equivalent to the search space covering initial design recommendations of Jones et al. (1998). We run each configuration for 10 random seeds. The performance is measured by evaluating the final policy induced by the configuration on a dedicated evaluation environment with a different random seed. We collect 128 episodes and calculate the average undiscounted cumulative reward, i.e. the return. This dataset can be found on GitHub as well as on Huggingface: `https://huggingface.co/datasets/autorl-org/arlbench`.

---

**Algorithm 1:** Subset Selection

---

**Input:** subset-size C, performance scores $p_\lambda^e$ for env. $e \in \mathcal{E}$ and config. $\lambda \in \Lambda$
**Output:** most predictive subset $\mathcal{I}^* = \{e_1^*, \cdots, e_k^*\}$
$\mathcal{I}_{\text{best}} = \emptyset$, $d_{\text{best}} = \infty$;
$(\overline{p}_\lambda^{\mathcal{E}})_{\lambda \in \Lambda} := (\frac{1}{|\mathcal{E}|} \cdot \sum_{e \in \mathcal{E}} p_\lambda^e)_{\lambda \in \Lambda}$;
**for** $\mathcal{I} = \{e_1, \cdots, e_k\} \subset \mathcal{E}$ **do**
   fit linear regression $f(p_\lambda^{e_1}, \cdots, p_\lambda^{e_k}) \mapsto \hat{p}_\lambda^{\mathcal{E}}$ to predict $\overline{p}_\lambda^{\mathcal{E}}$ for all $\lambda \in \Lambda$;
   $d_{\mathcal{I}} = 1 - \rho(f(p_\lambda^{e_1}, \cdots, p_\lambda^{e_k})_{\lambda \in \Lambda}, (\overline{p}_\lambda^{\mathcal{E}})_{\lambda \in \Lambda})$;
   **if** $d_{\mathcal{I}} < d_{best}$ **then**
      $\mathcal{I}_{\text{best}} = \mathcal{I}$; $d_{\text{best}} = d_{\mathcal{I}}$;

**return** $\mathcal{I}_{\text{best}}$;

---

## 4.2 Subset Selection

Based on the collected evaluation rewards, we aim to find a subset of environments on which to evaluate an AutoRL method. Due to discrete and continuous action spaces, the algorithms differ in the environments they are compatible with. Therefore, we perform this selection for each algorithm individually. The set of environments for PPO contains all 21 evaluated environments, while DQN is limited to discrete action spaces (13 environments), and SAC only supports continuous action spaces (8 environments). The full sets of environments per algorithm are listed in Appendix D. Details on the subset selection are stated in Appendix G.1.

**Finding an optimal subset.** For selecting an optimal subset, we use a method similar to Aitchison et al. (2023). Let $\Lambda$ be the set of hyperparameter configurations for an algorithm and $\mathcal{E}$ the corresponding set of environments. For each evaluated hyperparameter configuration $\lambda \in \Lambda$ and environment $e \in \mathcal{E}$, we are given a performance score $p_\lambda^e$. We define $\overline{p}_\lambda^{\mathcal{E}} := \frac{1}{|\mathcal{E}|} \cdot \sum_{e \in \mathcal{E}} p_\lambda^e$ as the average score of a configuration $\lambda$ across all environments. Given a subset of environments $\mathcal{I} \subset \mathcal{E}$ of size $C \in \mathbb{N}$, we use a linear regression model $f$ to predict $\overline{p}_\lambda^{\mathcal{E}}$ from the scores $p_\lambda^e$ for all $e \in \mathcal{I}$, i.e., $\hat{p}_\lambda^{\mathcal{E}} := f(p_\lambda^{e_1}, \cdots, p_\lambda^{e_C})$. An optimal subset $\mathcal{I}^*$ of size C is defined as

$$\mathcal{I}^* \in \operatorname*{arg\,min}_{\mathcal{I} = \{e_1, \cdots, e_C\} \subset \mathcal{E}} d(\hat{p}^{\mathcal{E}}, \overline{p}^{\mathcal{E}}) \text{ with } \hat{p}^{\mathcal{E}} = (\hat{p}_\lambda^{\mathcal{E}})_{\lambda \in \Lambda} \text{ and } \hat{p}_\lambda^{\mathcal{E}} = f(p_\lambda^{e_1}, \cdots, p_\lambda^{e_C}), \qquad (1)$$

where $d$ is a distance metric between the predicted and target hyperparameter landscapes, i.e., the vector of predicted scores $\hat{p}^{\mathcal{E}} = (\hat{p}_\lambda^{\mathcal{E}})_{\lambda \in \Lambda}$ and the vector of target scores $\overline{p}^{\mathcal{E}} = (\overline{p}_\lambda^{\mathcal{E}})_{\lambda \in \Lambda}$ spanning across the configurations $\lambda \in \Lambda$. The performance attained on the subset then provides the best approximation of the performance across all environments for subsets of size $C$. The pseudocode for computing the best subset of size $k$ is shown in Algorithm 1. Note that the environment selection does not take HPO behavior into account. We could perform the subselection to approximate HPO results directly. However, the performance discrepancies we see for HPO methods in the literature (Eimer et al., 2023; Shala et al., 2024) suggest that we do not yet know how to best apply HPO methods to RL. Therefore, we currently lack reliable methodologies to obtain the necessary data allowing us to infer direct relationships between environments and performance of HPO methods.

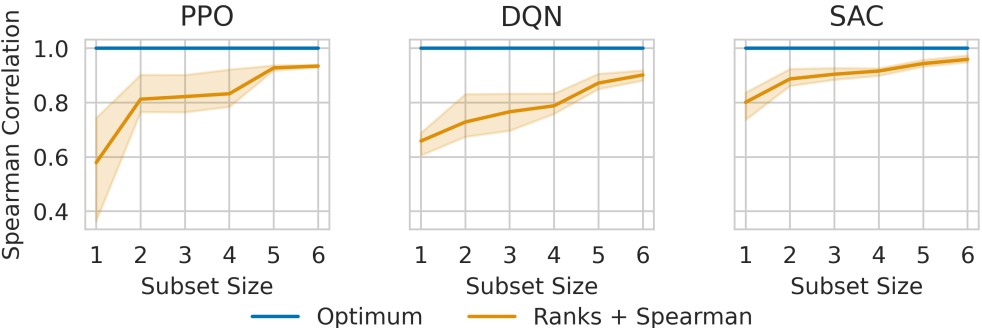

Figure 3: Comparison of the Spearman correlation for different subset sizes with confidence intervals from 5-fold cross-validation on the configurations.

**Selection Strategy.** Although reward scales vary drastically across environments, we lack the human expert scores (Aitchison et al., 2023) to normalize returns per environment. Instead, we apply a rank-based normalization method to obtain the performances $p^e_\lambda$. For an environment $e$, we train policies using each hyperparameter configuration $\lambda$ across 10 different random seeds and evaluate each resulting policy to obtain its mean return. The performance $p^e_\lambda$ of a configuration $\lambda$ is then determined by the average rank with respect to the mean return over its 10 random seeds when compared to all other configurations within the same environment $e$. Now, we can fit a linear model to predict the average ranks across the full set, given the ranks on the subset. We use the Spearman correlation coefficient $\rho_p$ as a similarity metric, leading to $d(\hat{p}^{\mathcal{E}}_\lambda, \overline{p}^{\mathcal{E}}_\lambda) := 1 - \rho_p(\hat{p}^{\mathcal{E}}_\lambda, \overline{p}^{\mathcal{E}}_\lambda)$ in Equation 1. Our choice of $\rho_p$ is motivated by our interest in capturing relationships between two return distributions robustly by focusing on relative rankings rather than exact values.

Figure 3 shows the Spearman correlation coefficient for the top three subsets of different sizes with confidence intervals computed using 5-fold cross-validation on the configurations. The results are fairly consistent for all algorithms except for very small subsets for PPO. Furthermore, they exhibit a high correlation to the full environment set, even when considering only a few environments. Based on these correlations, we select five environments for PPO and DQN from their respective full sets of 21 and 13 environments. The selected PPO subset shows a correlation of 0.95, while the DQN subset has a correlation of 0.92. For SAC, we select a subset of four environments, achieving a correlation of 0.94 with the full set of eight environments. At any point during training, the correlation between subset and full-set returns exceeds 0.9, making the subsets independent of training budget. Further details can be found in Figures 4 and 19 as well as Appendix G.3. A single training on all environments in all three subsets takes around 2.93 GPU hours, compared to 7.12 GPU hours for the full set of environments, where the ALE environments are limited to Atari-5 in the full set.

### 4.3 Validating ARLBench

Having selected a subset per algorithm, we still need to ensure this subset is representative of the full environment set from an HPO perspective. To investigate this, we examine (i)

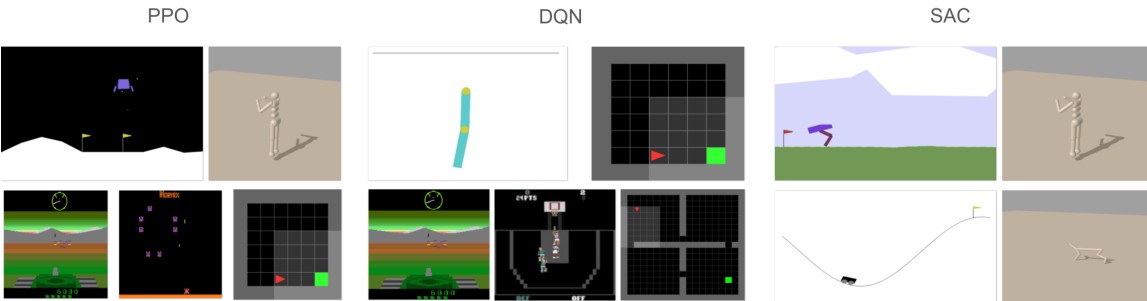

Figure 4: Selected set of representative environments per algorithm. For PPO, the discrete variant of LunarLander was selected.

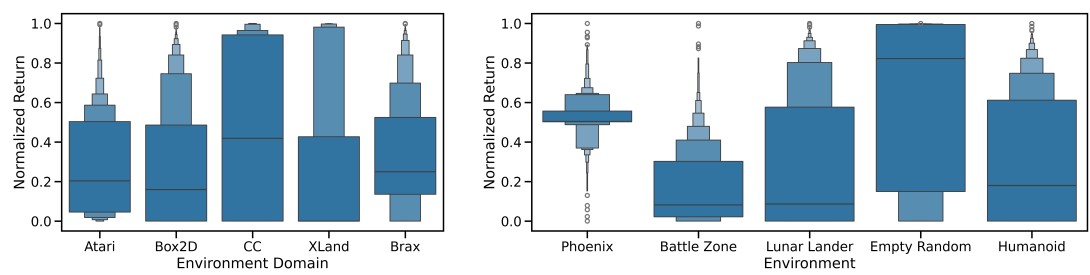

Figure 5: Comparison of the return distributions over hyperparameter configurations of PPO on all 21 environments (left) and the selected subset of 5 environments (right). For the same comparisons for DQN and SAC, see Appendix H.2.

the HPO landscape, in particular the return distributions and hyperparameter importance, and (ii) the performance of different HPO optimizers on the subset and full environment set. For most of the following analysis, we use DeepCAVE (Sass et al., 2022), as a monitoring package for HPO. We use 95% confidence intervals in our reported results as suggested by Agarwal et al. (2021).

**Comparing HPO Landscapes.** We first analyze the differences in HPO landscapes between the full environment set and our subset. This is especially important since we do not use HPO performance data for the selection but still want to ensure that HPO approaches will encounter the same overall landscape characteristics on the subsets as on all benchmarks. We argue that this yields the first insights into the consistency of HPO performance on the subset and full set of environments. To see if the overall RL algorithm performance changes, Figure 5 shows the distribution of returns in our random samples of PPO, normalized per domain by the performance scores seen in our pre-study. For the Box2D and Brax environments, we set fixed minimum scores of -200 and -2000, respectively, to mitigate artificially low performance caused by numerical instabilities. We see that the subset includes a diverse selection of return distributions: from a large bias of configurations towards the lower end of the performance spectrum (in BattleZone, LunarLander, and Humanoid), an even spread biased towards

| | PPO | Subset PPO | DQN | Subset DQN | SAC | Subset SAC |
|---|---|---|---|---|---|---|
| #HPs $\geq 5\%$ | 2.2 | 1.2 | 2.54 | 2.75 | 1.86 | 1.25 |
| #Interactions $\geq 5\%$ | 1.62 | 2.2 | 1.3 | 1.2 | 1.0 | 1.0 |

Table 1: Number of hyperparameters and hyperparameter interactions with over 5% importance on the full set and subset for each algorithm.

higher performance (in EmptyRandom) to a dense concentration of performances towards the middle (in Phoenix). Most environment domains show a similar trend to BattleZone, LunarLander, and EmptyRandom: there is a wide spread of configurations, with a bias towards low performances. Our subset thus captures this dominant trend as well as the tendency of XLand-Minigrid and Classic Control for a more even performance distribution. The Phoenix environment reflects the opposite behavior, ensuring that similar environments outside of the typical performance distribution are included in our subset. These different patterns in performance with regard to hyperparameter settings suggest that the selected subset is likely to test these variations in HPO behavior.

A large proportion of the behavior of RL algorithms regarding their hyperparameters is preserved in the subset selection (see Appendix H.2 for full results). In our fANOVA analysis (Hutter et al., 2014) (see Appendix H.3 for full results), we verify that the number of important hyperparameters stays consistent. For most algorithm-environment combinations, only two to four hyperparameters have an importance of at least 5%, though the specific important ones differ, similar to common observations in HPO (Bergstra and Bengio, 2012). Table 1 shows that the number of important hyperparameters and their interactions remain consistent in the subset, with the highest deviation being between 2.2 hyperparameters above 5% importance on average for PPO on the whole environment set and 1.2 on the subset. Our results, along with the observed similarities in return distributions, suggest that the main properties of the HPO landscapes are preserved in our subselection.

**Comparing HPO Optimizers.** To further validate the subset selection, we run four HPO optimizers with a budget of 32 full training runs each for all algorithms and environments. We use five runs, i.e., random seeds for each HPO optimizer and each configuration is evaluated on three random seeds during optimization, following recommendations by Eimer et al. (2023). We believe five seeds are a good compromise to obtain valid insights while accounting for the associated high computational demand. While statistically significant insights might require many more seeds, we believe five seeds are sufficient for obtaining preliminary insights into the compatibility of the full set of environments and our chosen subsets. To reflect the current range of HPO tools for RL, we select random search (RS; Bergstra and Bengio (2012)), PBT (Jaderberg et al., 2017), the Bayesian optimization tool SMAC (Lindauer et al., 2022) as well as SMAC in combination with the Hyperband scheduler (Li et al., 2017) (SMAC+HB). We compare the results on the subsets and the full set of environments.

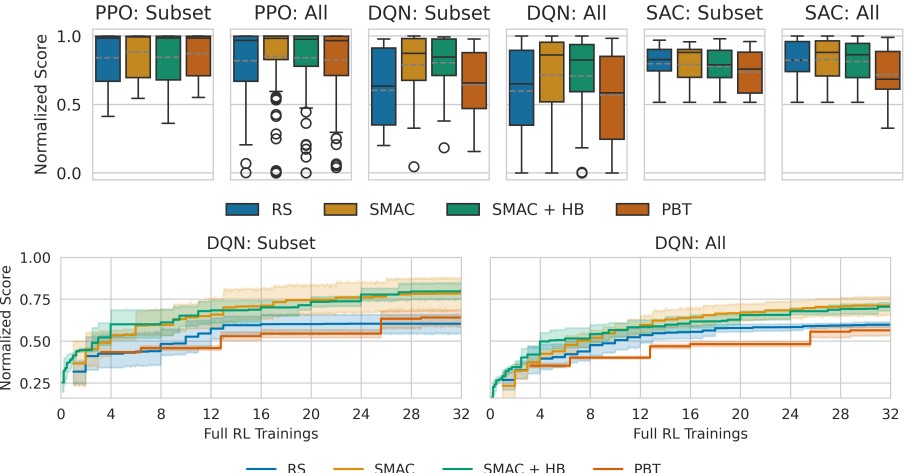

Figure 6: Comparison of the scores of our selected HPO methods on the subset and full environment (higher is better). **Top**: Performance distributions over optimizer runs and environments. Medians and means are visualized using black and dotted gray lines, respectively. **Bottom**: HPO anytime performance with 95% confidence intervals. See Appendices G.3 for PPO and SAC for details. We note that we do not consider inter-quartile means to prevent disregarding environments (top), especially since we are using only three optimizer runs (bottom).

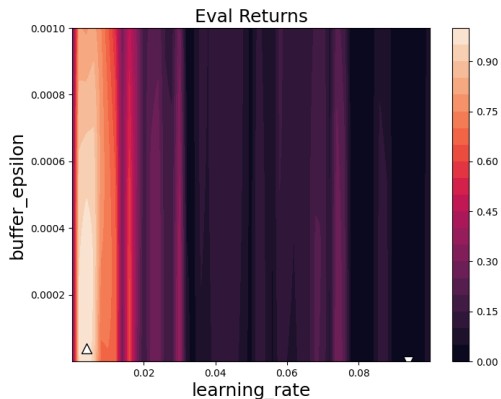

Figure 7: The hyperparameter landscape of DQN on CartPole-v1. Lighter is better, (mean performance over 10 seeds). Similar configurations perform very differently: *high returns occur next to almost failure modes.*

Figure 6 shows the HPO optimizer scores, normalized per domain by the performances seen in our pre-study, for each algorithm on the subset and all environments. The overall performance of each HPO optimizer is represented by the mean performance across all environments in the respective set of environments. We observe that the scores are distributed similarly between the full set and the subset on each algorithm for the final scores. Median and mean scores for all algorithms closely align with the respective scores of the subsets in terms of ranking.

For HPO anytime performance, the relative order remains consistent across both sets for SMAC and SMAC+HB compared to RS and PBT, with the only major difference being that SMAC, SMAC+HB, and PBT score higher on the subset; This, however, is due to merely a slight difference in

scores, which is still within the confidence intervals of SMAC and SMAC+HB as well as RS and PBT. Our analysis shows that overall, the best mean HPO optimizer performance is achieved by SMAC and SMAC+HB due to them being able to outperform RS and PBT on DQN. In many cases, however, we do not see a clear separation of performances, e.g. for RS and the SMAC variations on SAC or all optimizers on PPO. The overall trends we observe in the subsets and full sets of environments stay consistent, indicating the subsets provide a good approximation for the full set of environments.. In previous work (Eimer et al., 2023; Shala et al., 2024), multi-fidelity optimizers were shown to perform quite well, and PBT performed worst overall, which is consistent with our results. Another important factor is the inclusion of SAC where RS is especially strong. Even for PPO and DQN, however, it is striking how closely SMAC as a state-of-the-art HPO optimizer compares to RS, which typically performs worse than SMAC and other state-of-the-art HPO methods in standard supervised ML settings (Turner et al., 2021; Lindauer et al., 2022).

Looking at a partial hyperparameter landscape in Figure 7, we see a possible reason: this is far from the benign HPO landscapes Pushak and Hoos (2018) found for supervised learning. This shows that simply applying common HPO packages will not be sufficient to solve HPO for all RL tasks; a dedicated, specific effort is needed. We present further landscape plots in Appendix H.1, showing the contrast between benign and adverse landscapes we found during our experiments.

## 5 Limitations and Future Work

Due to the dimensions of complexity involved in this topic, including the computational expense and wealth of RL algorithms and environments, ARLBench has some limitations. First, we manually selected the underlying set of algorithms and environments from those used in the RL community at large. This gave rise to a focus on model-free learning in combination with base versions of PPO, DQN, and SAC. In the future, we will cover extensions to these algorithms, such as advanced types of replay strategies (Kapturowski et al., 2019), multi-step or exploration strategies (Amin et al., 2021; Pislar et al., 2022). Additionally, we would like to enable ARLBench to evaluate policy generalization, ensuring that optimized policies perform well in previously unseen environments (Kirk et al., 2023; Benjamins et al., 2023; Mohan et al., 2024; Benjamins et al., 2024). Further research on hyperparameter landscapes in RL (Mohan et al., 2023) can inform useful future additions.

Using a selected subset reduces computational costs but may increase variance due to the smaller sample size, requiring careful experimental design to ensure statistically significant results. Additionally, the selection of subsets could also consider training time, aiming for the most informative and the least costly subsets. Currently, we prioritize higher validation accuracy over reduced running time, even if one environment is slightly less important but significantly faster.

The computational cost itself remains a limitation of the benchmark. While our setting is much cheaper to evaluate and enables many more research groups to do thorough research on AutoRL, it is still by no means as cheap as surrogate or table lookups would be. Our highest priority is the flexibility of large configuration spaces and dynamic configuration, representing real-world HPO applications of RL; we do not see purely tabular benchmarks as an alternative in this exploratory phase of the field. Instead, we believe surrogate models (Eggensperger

et al., 2015, 2018; Zela et al., 2022) will be crucial for more efficient HPO in RL, though modeling the dynamic nature of HPO in surrogates remains an open challenge. Our published meta-dataset, the largest one for AutoRL to date, enables the first steps towards such dynamic surrogates.

Furthermore, there are additional elements of AutoRL research our benchmark does not yet fully support. We designed it to be future-oriented, with a benchmark structure that can, in principle, support second-order optimization methods, learning based on internal algorithmic aspects, such as losses and activation functions, or architecture search. However, we believe integrating these aspects into ARLBench first requires research into how state-based HPO in RL and NAS for RL should be approached. The same holds for concepts such as discovering RL algorithms (Co-Reyes et al., 2021; Jackson et al., 2024), where no standard interface exists for evaluating a learned algorithm. Nonetheless, our environment subsets can aid in the evaluation of these approaches. Integrating AutoRL for environment components, such as environment design (Jiang et al., 2021; Parker-Holder et al., 2022), into ARLBench poses a challenge because most RL environments do not inherently support these approaches. Improving the compatibility of the environment frameworks in ARLBench will facilitate the integration of these methods into the benchmark.

## 6 Conclusion

We propose a benchmark for HPO in RL that supports this emerging field of research by (i) providing a general, easily integrable and extensible way of evaluating various paradigms for HPO in RL; (ii) reducing computational costs with highly efficient implementations, while expanding the evaluation coverage of HPO methods by selecting informative environment subsets, achieving over 10 times the efficiency compared to standard frameworks; (iii) publishing a large set of performance data for future use in AutoRL research. Such a concerted effort is necessary to help the community work in a common direction and democratize AutoRL as a research field. While its set of algorithms and environments will evolve within the coming years, ARLBench is built to allow for easy extension, e.g., to AutoML paradigms such as NAS, which are currently underrepresented in RL. Therefore, ARLBench will catalyze the development of increasingly efficient HPO methods for RL that perform well across algorithms and environments.

BROADER IMPACT STATEMENT

By providing a publicly available benchmark and a rich dataset, ARLBench reduces computational and methodological barriers for researchers. This can allow for a more diverse and inclusive research community, driving innovation across various domains. The efficiency improvements embedded in ARLBench also address critical concerns about the environmental footprint of machine learning. With its significant reductions in compute time, ARLBench helps lower energy consumption and carbon emissions, supporting sustainable research practices. However, while ARLBench significantly reduces computational costs, the resource requirements remain considerable. Societal benefits from its use extend to practical applications of RL, where better-optimized algorithms could enable breakthroughs in fields such as robotics, healthcare, logistics, and energy management. Automation of RL design through benchmarks like ARLBench reduces reliance on domain-specific expertise, making advanced

RL technologies more accessible to practitioners and potentially accelerating progress in areas with direct societal benefits. However, RL also has applications in areas such as surveillance, autonomous weapon systems, and financial trading, where unchecked advancements raise ethical concerns and risks of societal harm. Moreover, reliance on benchmarks like ARLBench risks narrowing research focus to the tasks and domains represented in the benchmark. While ARLBench was designed to be diverse and representative, no benchmark can fully capture the range of challenges encountered in real-world RL tasks. This overemphasis on benchmark performance could lead to progress that is less generalizable or applicable to novel, real-world scenarios.

### Acknowledgments

We gratefully acknowledge computing resources provided by the NHR Center NHR4CES at RWTH Aachen University (p0021208). Further, the computing time provided on the high-performance computer Noctua2 at the NHR Center PC2 under the project hpc-prf-intexml. These are funded by the Federal Ministry of Education and Research and the state governments participating on the basis of the resolutions of the GWK for the national high performance computing at universities (www.nhr-verein.de/unsere-partner). Theresa Eimer acknowledges funding by the German Research Foundation (DFG) under LI 2801/10-1. Raghu Rajan acknowledges funding through the research network "Responsive and Scalable Learning for Robots Assisting Humans" (ReScaLe) of the University of Freiburg. The ReScaLe project is funded by the Carl Zeiss Foundation. This research was supported in part by an Alexander von Humboldt Professorship in AI held by Holger Hoos and by the "Demonstrations- und Transfernetzwerk KI in der Produktion (ProKI-Netz)" initiative, funded by the German Federal Ministry of Education and Research (BMBF, grant number 02P22A010).

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

## Appendix A. Dataset Description

Our dataset is hosted on HuggingFace for easy and continued access: `https://huggingface.co/datasets/autorl-org/arlbench`. See the page there for in-depth information on data value and distributions. The croissant meta-data can also be found here: `https://github.com/automl/arlbench/blob/experiments/croissant_metadata.json`. Everything needed to reproduce the data can be found in the *experiments* branch of our GitHub repository: `https://github.com/automl/arlbench/tree/experiments` It is intended to be used in continued research on AutoRL, e.g., by using it in warm-starting HPO optimizers, proposing novel analysis methods or meta-learning on it. The dataset is in CSV format, making it easily readable. We license it under the BSD-3 license.

## Appendix B. Reproducing Our Results

Below we describe our hardware setup and steps for reproducing our experiments.

### B.1 Execution Environment

To conduct the experiments detailed in this paper, we pooled various computing resources. Below, we describe the different hardware setups used for CPU and GPU-based training.

**CPU Jobs.** Compute nodes with CPUs of type AMD Milan 7763, 2.45 GHz, each 2x 64 cores, 128GB main memory

**GPU Jobs.**

V100 Cluster: Compute nodes with CPUs of type Intel Xeon Platinum 8160, 2.1 GHz, each 2x 24 cores, 180GB main memory. Each node comes with 16 GPUs of type NVIDIA V100-SXM2 with NVLink and 32 GB HBM2, 5120 CUDA cores, 640 Tensor cores, 128 GB main memory

A100 Cluster: Compute nodes with CPUs of type AMD Milan 7763, 2.45 GHz, each 2x 64 cores, 126GB main memory. Each node comes with 1 GPU of type NVIDIA A100 with NVLink and 40 GB HBM2, 6,912 CUDA cores, 432 Tensor cores, 16 GB main memory

H100 Cluster: Compute nodes with CPUs of type Intel Xeon 8468 Sapphire, 2.1 GHz, each 2x 48 cores, 512GB main memory. Each node comes with 4 GPUs of type NVIDIA H100 with NVLink and 96 GB HBM2e, 16,896 CUDA cores, 528 Tensor cores, 512 GB main memory

### B.2 Experiment Code

We provide code and runscripts for all of our dependencies in the *experiments* branch of our repository: `https://github.com/automl/arlbench/tree/experiments`.

All scripts relating to the dataset creation and HPO optimizer runs are in *runscripts*. For the performance over time plots, see *runtime_ comparison. rs_ data_ analysis* contains the analysis of the HPO landscapes. For the subset selection, see *subset_ selection*. The subset validation and performance over time plots for the HPO optimizers can be found in *subset_ validation*. Additionally, we provide all of our raw data in *results_ finished* with

*results_combined* containing dataset aggregates. Instructions for the usage of all of these can be found in the ReadMe file of that branch.

## Appendix C. Maintenance Plan

Following (Eggensperger et al., 2021) and (Pfisterer et al., 2022), we provide a maintenance plan for the future of ARLBench. For our feature roadmap, see: `https://github.com/orgs/automl/projects/17`

**Who Maintains.** ARLBench is being developed and maintained as a cooperation between the Institute of AI at the Leibniz University of Hannover and the chair for AI Methodology at the RWTH Aachen University.

**Contact.** Improvement requests, issues and questions can be asked via issue in our GitHub repository: `https://github.com/automl/arlbench`. The contact e-mails we provide can be used for the same purpose.

**Errata.** There are no errata.

**Library Updates.** We plan on updating the library with new features, specifically more extensive state features, more algorithms and added environment frameworks. We also welcome updates via external pull requests which we will test and integrate into ARLBench. Changes will be communicated via the changelog of our GitHub and PyPI releases.

**Support for Older Versions.** Older versions of ARLBench will continue to be available on PyPI and GitHub, but we will only provide limited support.

**Contributions.** Contributions to ARLBench from external parties are welcome in any form, be extensions to other environment frameworks, added algorithms or extensions of the core interface. We describe the contribution process in our documentation: `https://automl.github.io/arlbench/main/CONTRIBUTING.html`. These contributions are managed via GitHub pull requests.

**Dependencies.** All of our dependencies are listed here in the GitHub repository: `https://github.com/automl/arlbench/blob/main/pyproject.toml`.

## Appendix D. Overview of all Environments

Tables 2, 3 and 4 provide an overview of all environments we executed for a given RL algorithm, including the underlying framework used and the number of environment steps for training.

| Category | Framework | Name | #timesteps |
|---|---|---|---|
| ALE | Envpool | BattleZone-v5 | $10^7$ |
| ALE | Envpool | DoubleDunk-v5 | $10^7$ |
| ALE | Envpool | Phoenix-v5 | $10^7$ |
| ALE | Envpool | Qbert-v5 | $10^7$ |
| ALE | Envpool | NameThisGame-v5 | $10^7$ |
| Box2D | Envpool | LunarLander-v2 | $10^6$ |
| Box2D | Envpool | LunarLanderContinuous-v2 | $10^6$ |
| Box2D | Envpool | BipedalWalker-v3 | $10^6$ |
| Walker | Brax | Ant | $5 \cdot 10^7$ |
| Walker | Brax | HalfCheetah | $5 \cdot 10^7$ |
| Walker | Brax | Hopper | $5 \cdot 10^7$ |
| Walker | Brax | Humanoid | $5 \cdot 10^7$ |
| Classic Control | Gymnax | Acrobot-v1 | $10^6$ |
| Classic Control | Gymnax | CartPole-v1 | $10^5$ |
| Classic Control | Gymnax | MountainCarContinuous-v0 | $2 \cdot 10^4$ |
| Classic Control | Gymnax | MountainCar-v0 | $10^6$ |
| Classic Control | Gymnax | Pendulum-v1 | $10^5$ |
| XLand | XLand-Minigrid | MiniGrid-DoorKey-5x5 | $10^6$ |
| XLand | XLand-Minigrid | MiniGrid-EmptyRandom-5x5 | $10^5$ |
| XLand | XLand-Minigrid | MiniGrid-FourRooms | $10^6$ |
| XLand | XLand-Minigrid | MiniGrid-Unlock | $10^6$ |

Table 2: ARLBench Environments for PPO with their respective training timesteps.

| Category | Framework | Name | #timesteps |
|---|---|---|---|
| ALE | Envpool | BattleZone-v5 | $10^7$ |
| ALE | Envpool | DoubleDunk-v5 | $10^7$ |
| ALE | Envpool | Phoenix-v5 | $10^7$ |
| ALE | Envpool | Qbert-v5 | $10^7$ |
| ALE | Envpool | NameThisGame-v5 | $10^7$ |
| Box2D | Envpool | LunarLander-v2 | $10^6$ |
| Classic Control | Gymnax | Acrobot-v1 | $10^5$ |
| Classic Control | Gymnax | CartPole-v1 | $5 \cdot 10^4$ |
| Classic Control | Gymnax | MountainCar-v0 | $12 \cdot 10^4$ |
| XLand | XLand-Minigrid | MiniGrid-DoorKey-5x5 | $10^6$ |
| XLand | XLand-Minigrid | MiniGrid-EmptyRandom-5x5 | $10^5$ |
| XLand | XLand-Minigrid | MiniGrid-FourRooms | $10^6$ |
| XLand | XLand-Minigrid | MiniGrid-Unlock | $10^6$ |

Table 3: ARLBench Environments for DQN with their respective training timesteps.

| Category | Framework | Name | #timesteps |
|---|---|---|---|
| Box2D | Envpool | LunarLanderContinuous-v2 | $5 \cdot 10^5$ |
| Box2D | Envpool | BipedalWalker-v2 | $5 \cdot 10^5$ |
| Walker | Brax | Ant | $5 \cdot 10^6$ |
| Walker | Brax | HalfCheetah | $5 \cdot 10^6$ |
| Walker | Brax | Hopper | $5 \cdot 10^6$ |
| Walker | Brax | Humanoid | $5 \cdot 10^6$ |
| Classic Control | Gymnax | MountainCarContinuous-v0 | $5 \cdot 10^4$ |
| Classic Control | Gymnax | Pendulum-v1 | $2 \cdot 10^4$ |

Table 4: ARLBench Environments for SAC with their respective training timesteps.

## Appendix E. Performance Comparisons with Other RL Frameworks

To validate the correctness of our implementations beyond unit testing, we compare their performance on a range of environments to established RL frameworks in terms of achieved return and running time. Additionally, running time comparisons for an HPO method of 32 RL runs, using 10 seeds each on the full environment set and our subsets between ARLBench and SB3 for each environment category are shown in Figures 8, 9, 10, 11, and12.

Figure 13 compares the resulting learning curves between ARLBench and SB3. For Brax environments, we use the default implementations for PPO and SAC included in Brax as our baselines, since SB3 performed significantly worse compared to the Brax algorithm implementations. In most of our tests, we observe very similar behavior with the other frameworks, with ARLBench outperforming the other two times and showing comparable learning curves for all other experiments. In the case of DQN, where SB3 performed better on CartPole and worse on Pong, the results of SB3 look noisy, possibly causing this discrepancy in both directions. SB3 also outperforms ARLbench on Pendulum, though this difference is fairly slight. For PPO on Ant, ARLBench performs quite a bit better than the Brax default agent, though their performances of SAC are the same. Inconsistencies in learning curves can be due to differences in the implementations of algorithms[1] and environments (Voelcker et al., 2024). Overall, this shows that our algorithms perform on par with other commonly used implementations.

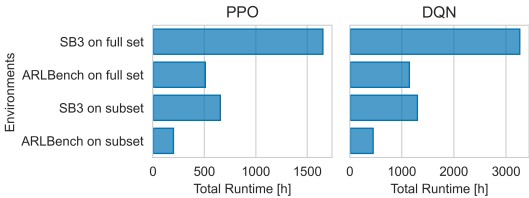

Figure 8: Running time comparison between ARLBench and SB3 for ALE. JAX-related speedup factors are 3.21 for PPO and 2.83 for DQN. Total speedup factors of the ARLBench subset compared to the full set of environments in SB3 are 8.03 for PPO and 7.08 for DQN. Note: As ALE environments have discrete action spaces, SAC is left out in this figure.

---

1. https://iclr-blog-track.github.io/2022/03/25/ppo-implementation-details/

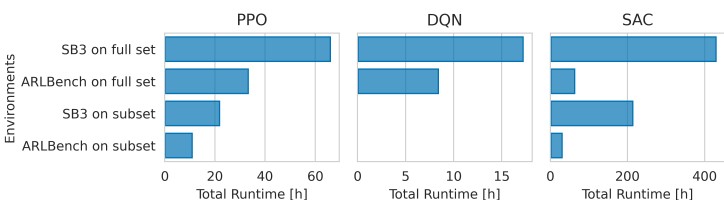

Figure 9: Running time comparison between ARLBench and SB3 for Box2D. JAX-related speedup factors are 1.97 for PPO, 2.04 for DQN, and 6.64 for SAC. Total speedup factors of the ARLBench subset compared to the full set of environments in SB3 are 5.92 for PPO and 13.27 for SAC. As no Box2D environment is part of the DQN subset, there is no total speedup factor for DQN.

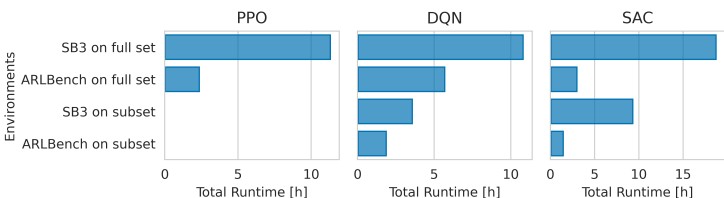

Figure 10: Running time comparison between ARLBench and SB3 for Classic Control. JAX-related speedup factors are 4.72 for PPO, 1.89 for DQN, and 6.10 for SAC. Total speedup factors of the ARLBench subset compared to the full set of environments in SB3 are 5.68 for DQN and 12.2 for SAC. As no Classic Control environment are part of the PPO subset, there is no total speedup factor for PPO.

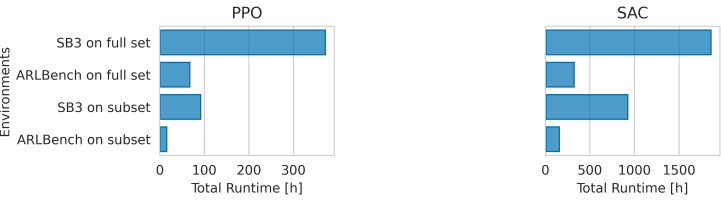

Figure 11: Running time comparison between ARLBench and SB3 for Brax. JAX-related speedup factors are 5.4 for PPO and 5.64 for SAC. Total speedup factors of the ARLBench subset compared to the full set of environments in SB3 are 21.62 for PPO, and 11.28 for SAC. Note: As Brax environments have continuious action spaces DQN is left out in this figure.

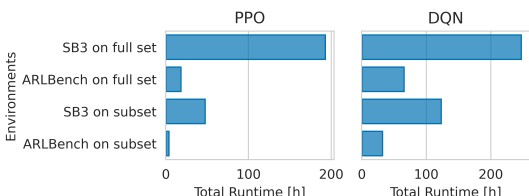

Figure 12: Running time comparison between ARLBench and SB3 for XLand-Minigrid. JAX-related speedup factors are 10.02 for PPO and 3.72 for DQN. Total speedup factors of the ARLBench subset compared to the full set of environments in SB3 are 40.07 for PPO and 7.42 for DQN. Note: As ALE environments have discrete action spaces SAC is left out in this figure.

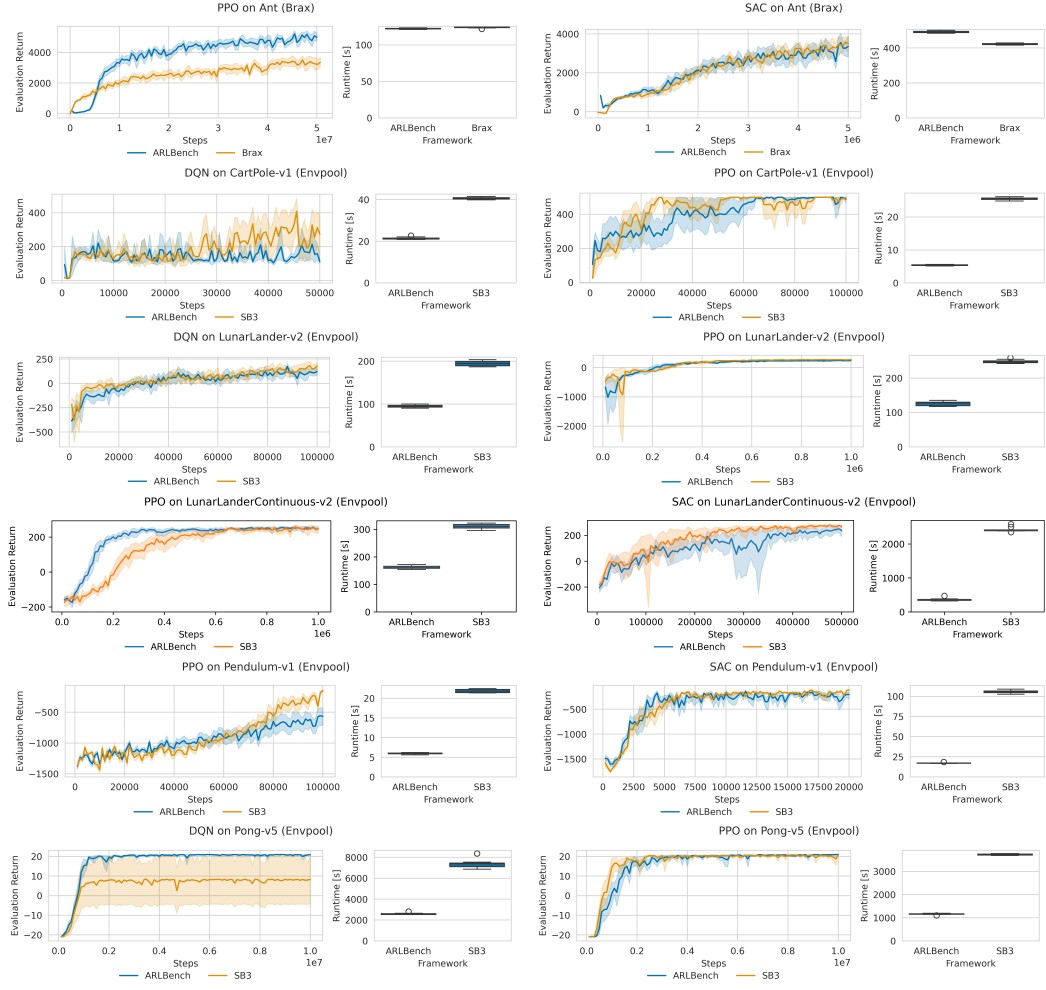

Figure 13: Performance comparisons of ARLBench, SB3 and the Brax agents.

Table 5 show the speedups we achieve in terms of running time over SB3 on all subsets while Tables 6, 7 and 8 list the same for each environment individually. As already discussed, we see a consistently large speedup, most pronounced for the Brax walkers with a factor of 8.57 for PPO and 10.67 for SAC. The lowest speedups we observe are still close to a factor of 2: 1.89 for DQN CartPole as well as 1.91 and 1.97 respectively for PPO LunarLander and LunarLanderContinuous.

| Algorithm | Set | ARLBench | SB3 | Speedup |
|---|---|---|---|---|
| PPO | All | 2h | 7.18h | 3.59 |
| PPO | Subset | 0.74h | 2.58h | 3.48 |
| DQN | All | 3.87h | 11.10h | 2.87 |
| DQN | Subset | 1.55h | 4.49h | 2.89 |
| SAC | All | 1.25h | 7.23h | 5.78 |
| SAC | Subset | 0.62h | 3.61h | 5.82 |
| **Sum** | All | 7.12h | 25.51h | 3.58 |
| **Sum** | Subset | 2.91h | 10.68h | 3.67 |

Table 5: Running time comparisons for a single RL training between ARLBench and SB3 on the set of all environments and the selected subset. The numbers are based on the results in Tables 6, 7, and 8. For each environment category, we use the running times from the experiments to estimate the overall running time for this category.

| Category | Framework | Name | ARLBench | SB3 | Speedup |
|---|---|---|---|---|---|
| Classic Control | Envpool | CartPole-v1 | 5.42s | 25.54s | 4.72 |
| Classic Control | Envpool | Pendulum-v1 | 5.98s | 21.87s | 3.66 |
| Box2D | Envpool | LunarLander-v2 | 125.87s | 248.54s | 1.97 |
| Box2D | Envpool | LunarLanderContinuous-v2 | 162.77s | 311.25s | 1.91 |
| XLand | XLand-Minigrid | Minigrid-DoorKey-5x5 | 54.38s | 544.71s | 10.01 |
| ALE | Envpool | Pong-v5 | 1161.11s | 3728.58s | 3.21 |
| Walker | Envpool | Ant | 194.09s | 1048.84s | |
| Walker | Brax | Ant | 122.28s | 1048.84s* | 8.57 |
| | | | | **Average** | 4.86 |

Table 6: Speedup of ARLBench PPO compared to SB3 on different environments. *Note: Since SB3 is not compatible with Brax without manual interface adaptation, we compare the results of MuJoCo + SB3 and Brax + ARLBench.

| Category | Framework | Name | ARLBench | SB3 | Speedup |
|---|---|---|---|---|---|
| Classic Control | Envpool | CartPole-v1 | 21.5s | 40.68s | 1.89 |
| Box2D | Envpool | LunarLander-v2 | 95.27s | 194.61s | 2.04 |
| XLand | XLand-Minigrid | Minigrid-DoorKey-5x5 | 187.73s | 697.64s | 3.71 |
| ALE | Envpool | Pong-v5 | 2602.69s | 7373.40s | 2.83 |
| | | | | **Average** | 2.15 |

Table 7: Speedup of ARLBench DQN compared to SB3 on different environments.

| Category | Framework | Name | ARLBench | SB3 | Speedup |
|---|---|---|---|---|---|
| Classic Control | Envpool | Pendulum-v1 | 17.32s | 105.67s | 6.10 |
| Box2D | Envpool | LunarLanderContinuous-v2 | 365.45s | 2425.04s | 6.64 |
| Walker | Envpool | Ant | 930.06s | 5245.17s | |
| Walker | Brax | Ant | 491.70s | 5245.17s* | 10.67 |
| | | | | **Average** | 7.80 |

Table 8: Speedup of ARLBench SAC compared to SB3 on different environments. *Note: Since SB3 is not compatible with Brax without manual interface adaptation, we compare the results of MuJoCo + SB3 and Brax + ARLBench.

## Appendix F. Algorithm Search Spaces

For all algorithms, we used extensive search spaces covering almost all hyperparameters that are commonly optimized. The search spaces for PPO, DQN and SAC are presented in Table 9, 10 and 11 respectively. We choose not to optimize hyperparameters that impact the running time of training to keep the computational resources constant for each training run. The default values for these hyperparameters for each environment domain have been inferred from stable-baselines3 zoo (Raffin, 2020) and the hyperparameter sweeps of Brax (Freeman et al., 2021) and are shown in Tables 9, 10 and 11 accordingly. The search space for the batch sizes was set to one power of two below and above its baseline value.

| Hyperparameter | Box2D | XLand | ALE | CC | Brax |
|---|---|---|---|---|---|
| batch size | $\{32, 64, 128\}$ | | $\{128, 256, 512\}$ | | $\{512, 1024, 2048\}$ |
| number of environments | 16 | | 8 | | 2048 |
| number of steps | 1024 | 32 | 128 | 32 | 512 |
| update epochs | 4 | 10 | | 4 | |
| learning rate | $\log([10^{-6}, 10^{-1}])$ | | | | |
| entropy coefficient | $[0.0, 0.5]$ | | | | |
| gae lambda | $[0.8, 0.9999]$ | | | | |
| policy clipping | $[0.0, 0.5]$ | | | | |
| value clipping | $[0.0, 0.5]$ | | | | |
| normalize advantages | $\{Yes, No\}$ | | | | |
| value function coefficient | $[0.0, 1.0]$ | | | | |
| max gradient norm | $[0.0, 1.0]$ | | | | |

Table 9: The hyperparameter search space for PPO. To keep the computational costs feasible, we choose not to optimize the number of steps per epoch and update epochs.

| Hyperparameter | ALE | Box2D | CC | XLand |
|---|---|---|---|---|
| batch size | $\{16, 32, 64\}$ | $\{64, 128, 256\}$ | | $\{32, 64, 128\}$ |
| number of environments | 8 | 4 | 1 | 4 |
| buffer priority sampling | | $\{$Yes, No$\}$ | | |
| buffer $\alpha$ | | $[0.01, 1.0]$ | | |
| buffer $\beta$ | | $[0.01, 1.0]$ | | |
| buffer $\epsilon$ | | $\log([10^{-7}, 10^{-3}])$ | | |
| buffer size | | $[1024, 10^6]$ | | |
| initial epsilon | | $[0.5, 1.0]$ | | |
| target epsilon | | $[0.001, 0.2]$ | | |
| learning rate | | $\log([10^{-6}, 10^{-1}])$ | | |
| learning starts | | $[1, 2048]$ | | |
| use target network | | $\{$Yes, No$\}$ | | |
| target update interval | | $[1, 2000]$ | | |

Table 10: The hyperparameter search space for DQN. The target update interval is a conditional hyperparameter that is only optimized when a target network is used. Similarly, buffer $\alpha$, $\beta$ and $\epsilon$ are only optimized when priority sampling is used. If the number of training steps is smaller than the upper limit of the buffer size, the buffer size limit is reduced accordingly.

| Hyperparameter | Box2D | CC | Brax |
|---|---|---|---|
| batch size | $\{128, 256, 512\}$ | $\{256, 512, 1024\}$ | $\{512, 1024, 2048\}$ |
| number of environments | 1 | 1 | 64 |
| buffer priority sampling | | $\{$Yes, No$\}$ | |
| buffer $\alpha$ | | $[0.01, 1.0]$ | |
| buffer $\beta$ | | $[0.01, 1.0]$ | |
| buffer $\epsilon$ | | $\log([10^{-7}, 10^{-3}])$ | |
| buffer size | | $[1024, 10^6]$ | |
| learning rate | | $\log([10^{-6}, 10^{-1}])$ | |
| learning starts | | $[1, 2048]$ | |
| use target network | | $\{$Yes, No$\}$ | |
| target network $\tau$ | | $[0.01, 1.0]$ | |
| reward scale | | $\log([0.1, 10])$ | |

Table 11: The hyperparameter search space for SAC. The target network hyperparameter $\tau$ is a conditional parameter that is only optimized when a target network is used. Similarly, buffer $\alpha$, $\beta$ and $\epsilon$ are only optimized when priority sampling is used. If the number of training steps is smaller than the upper limit of the buffer size, the buffer size limit is reduced accordingly.

## Appendix G. Subset Selection

We provide additional information on the subset selection in the form of explanations, alternative selection methods, and a more detailed look into the results, including environment weights.

### G.1 Performance Metrics and Rank-Based Normalization

We select the subset based on the hyperparameter landscapes obtained through Sobol sampling. For each randomly sampled hyperparameter configuration, the RL algorithm is trained and evaluated on a separate evaluation environment. As an evaluation metric, we

collect the undiscounted cumulative episode rewards, i.e., return of 128 episodes and calculate the mean. The mean return for environment $e \in \mathcal{E}$ and hyperparameter configuration $\lambda \in \Lambda$ is denoted as $r^e_\lambda$ and calculated as

$$r^e_\lambda = \mathbb{E}_{s \sim \mathcal{S}} \left[ \frac{1}{128} \cdot \sum_{i=1}^{128} \sum_{t=1}^{T} R_t^{(i,s)} \right] \quad (2)$$

where $\mathcal{S}$ is the set of 10 random seeds, $T$ is the number of steps in the $i$-th evaluation episode, and $R_t$ corresponds to the reward at time step $t$ in the $i$-th episode for seed $s$. As reward ranges differ across environments, we have to apply normalization to compare the corresponding returns. However, normalization based on human expert scores is not possible as done by Aitchison et al. (2023) for the selection of Atari-5. We apply rank-based normalization to compare the returns of different environments. By ranking the returns $r^e_\lambda$ of all configurations $\lambda \in \Lambda$ for a given environment $e$, with higher returns corresponding to higher ranks, and normalizing these ranks to the interval $[0,1]$, we obtain the performance scores $p^e_\lambda$. The performance score $p^e_\lambda$ for each configuration $\lambda$ in environment $e$ is given by:

$$p^e_\lambda = \frac{\text{rank}(r^e_\lambda) - \min_{\lambda' \in \Lambda} \text{rank}(r^e_{\lambda'})}{\max_{\lambda' \in \Lambda} \text{rank}(r^e_{\lambda'}) - \min_{\lambda' \in \Lambda} \text{rank}(r^e_{\lambda'})}, \quad (3)$$

where $\text{rank}(r^e_\lambda)$ denotes the rank of the return $r^e_\lambda$ among all returns in environment $e$.

For the regression model, we used the *LinearRegression* class from the *scikit-learn*[2] package. This relies on the ordinary least squares method for fitting, which is invariant to permutation of features, i.e., environments.

## G.2  Alternative Selection Methods

In addition to our chosen method of rank-based normalization in combination with the Spearman correlation as a distance metric, we compare alternative normalization methods as well as MSE for the distance. Figure 14 shows the validation error of different combinations while Figure 15 shows the resulting Spearman correlation to the full environment set. While MSE might produce a good validation error, the resulting correlation is significantly worse than using the Spearman correlation for the distance. Min-max normalization performs slightly worse than rank normalization for the validation error. Therefore we chose rank-based normalization with Spearman correlation for our subset selection.

---

2. `https://scikit-learn.org`

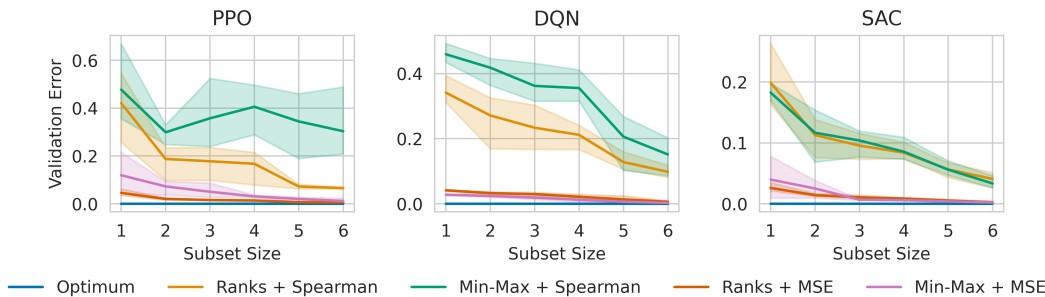

Figure 14: Validation error across subset sizes for min-max and rank methods using MSE and Spearman error functions.

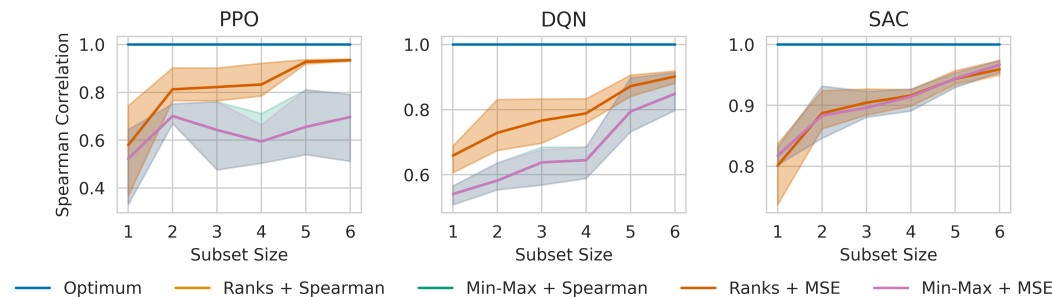

Figure 15: Validation error across subset sizes for min-max and rank methods using MSE and Spearman error functions. Please note, that not all lines in this plot are visible due to overlaps. The reason is that the approaches Ranks + Spearman and Ranks + MSE as well as Min-Max + MSE and Min-Max Spearman each results in the exact same Spearman correlation and thus are not distinguishable in the plot.

## G.3 Extended Subset Results

In addition to the environments in the subsets, we also provide the exact weights for each environment in the subsets in Table 12. Furthermore, Figures 16, 17, and 18 show the optimization-over-time results for PPO, DQN, and SAC. The results for each category of environments as well as for individual environments can be found in `https://github.com/automl/arlbench/tree/experiments/plots/subset_validation/optimizer_runs`.

| Algorithm | Environments (with predicted weights) | $\rho_s$ |
|---|---|---|
| PPO | $0.21\times$ LunarLander, $0.21\times$ Humanoid, $0.18\times$ BattleZone, $0.12\times$ Phoenix, $0.23\times$ MiniGrid-EmptyRandom | 0.96 |
| DQN | $0.33\times$ Acrobot, $0.11\times$ BattleZone, $0.22\times$ DoubleDunk $0.12\times$ MiniGrid-FourRooms, $0.18\times$ MiniGrid-EmptyRandom | 0.96 |
| SAC | $0.32\times$ BipedalWalker, $0.31\times$ HalfCheetah, $0.15\times$ Hopper, $0.19\times$ MountainCarContinuous | 0.97 |

Table 12: The environment subsets selected for each algorithm with their Spearman correlation to the full environment set. In addition, we depict the weighting of the score of each environment in the fitted linear regression function.

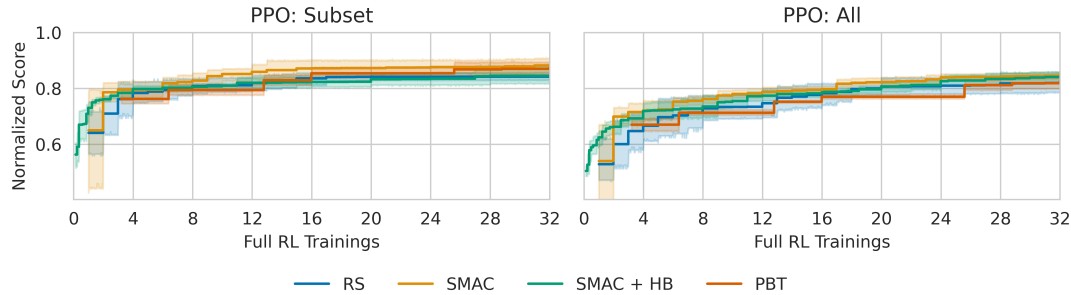

Figure 16: Anytime performance of the HPO methods for PPO.

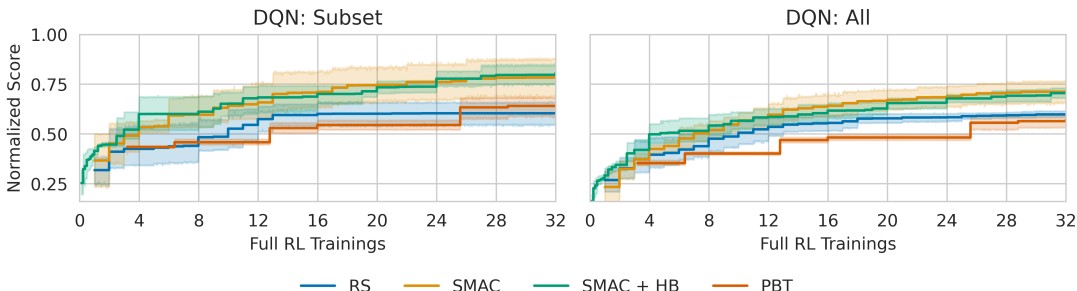

Figure 17: Anytime performance of the HPO methods for DQN.

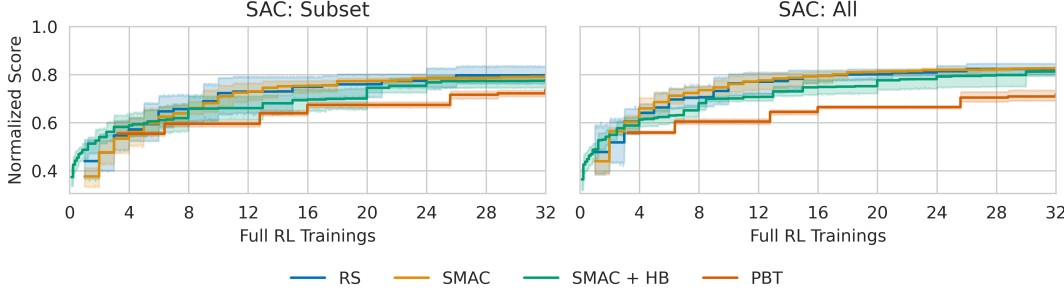

Figure 18: Anytime performance of the HPO methods for SAC.

To evaluate the robustness of ARLBench subsets to changes in learning curves, we analyzed the Spearman rank correlation between configuration performances on the subsets and the full sets of environments after different training steps. As shown in Figure 19, the correlation remains consistently higher than 0.85 across algorithms and throughout the entire range of normalized training steps. This result highlights that the relative ranking of configurations is stable, even during the early phases of training.

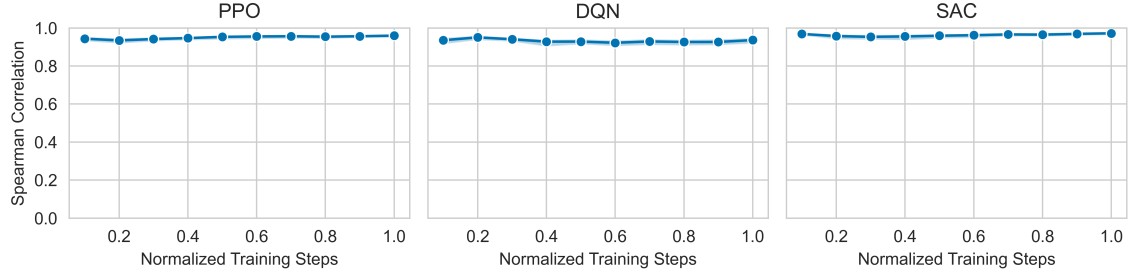

Figure 19: Spearman rank correlation between configuration evaluation returns on the subset and full set across normalized training steps for PPO, DQN, and SAC. The correlation is computed using 100 bootstrapped samples of 256 configurations each, with performance aggregated as the mean across all environments for each budget. Error bars represent the 95% confidence intervals across bootstrapped samples.

To illustrate the potential reason for this consistency across training steps, Figures 20, 21 and 22 shows the learning curves configurations at the 0%, 25%, 50%, 75% and 100% performance percentiles of each subset. While there is of course variation over time, these trends look quite benign in our plots, showing e.g. no sudden desctructive performance drops. Therefore, the variation between environments rather than the budget is most distinctive.

This is confirmed if we look at the overall performance distribution every 10% of training steps for the subsets and full environment sets (see Figure 23). We observe the median performance rising in a similar, gradual trajectory for the full set and the subset, with slightly wider performance distribution over time. There are outliers, especially for SAC, and the performance distribution for the full sets of PPO and DQN is narrower, likely in part due to the much larger amount of data. The majority of runs, however, seem to follow a more or less predictable pattern over time. We believe this is the reason our subsets generalize fairly well across training budgets.

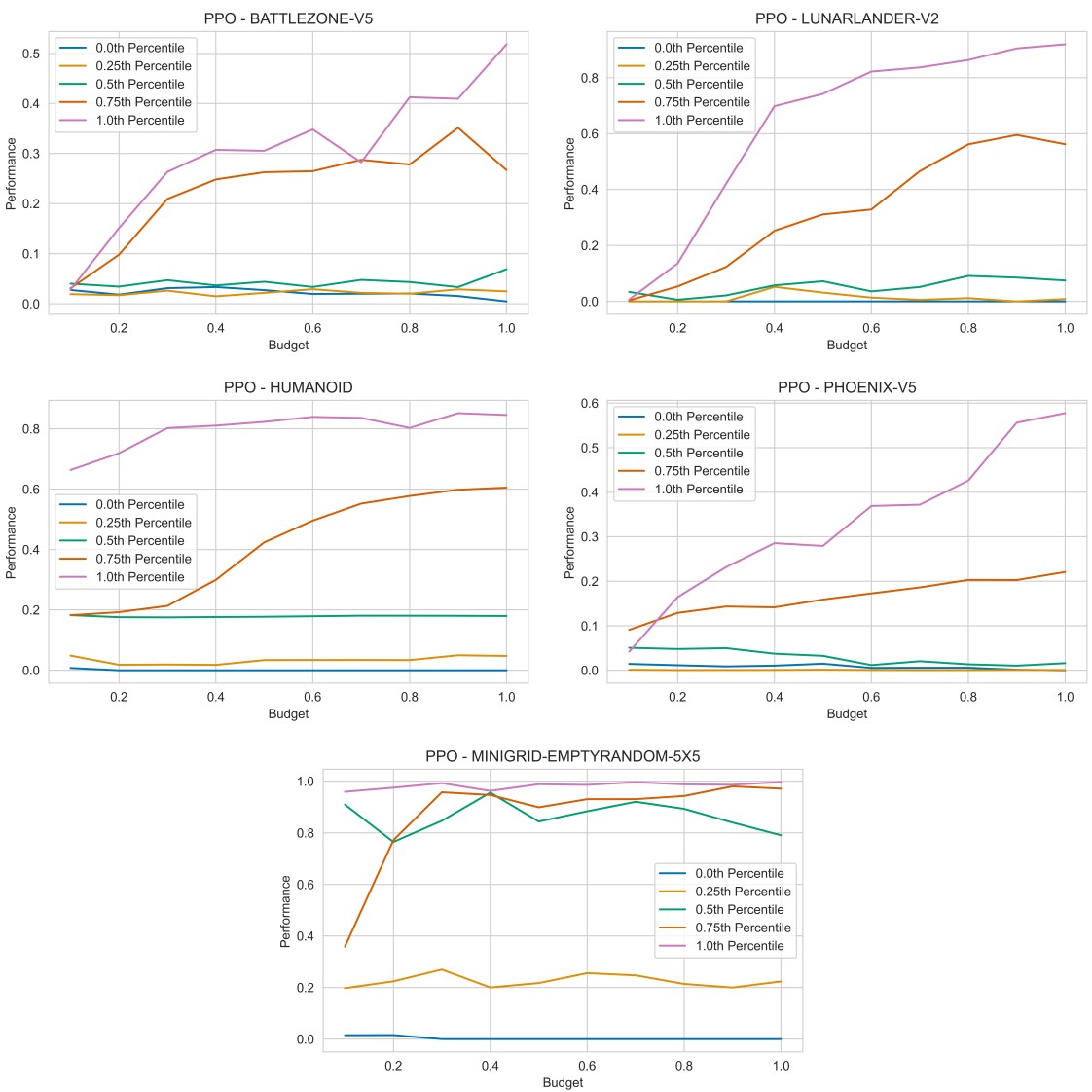

Figure 20: Learning curves of configurations at the 0th, 25th, 50th, 75th and 100th performance percentiles in the collected hyperparameter landscapes for the PPO subset.

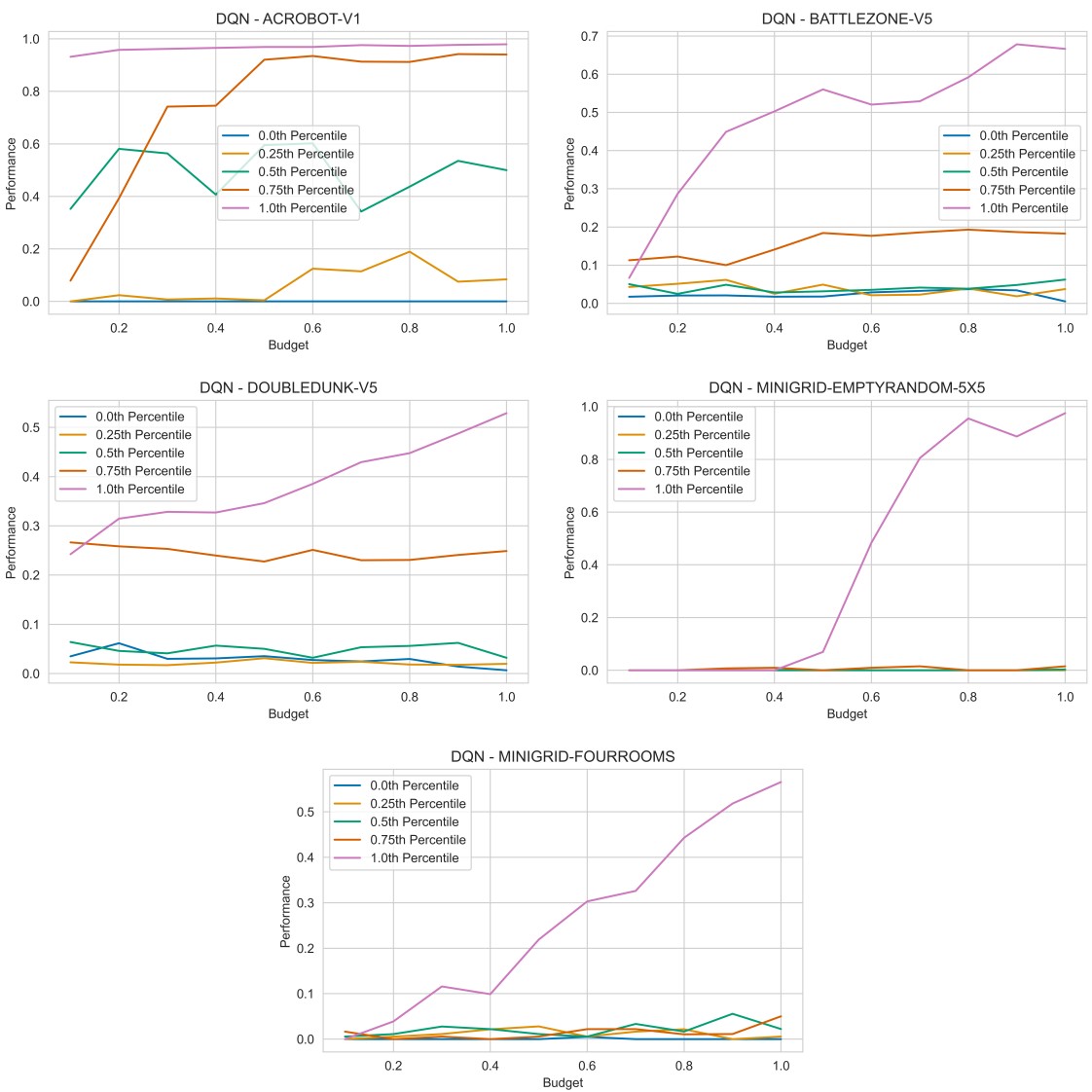

Figure 21: Learning curves of configurations at the 0th, 25th, 50th, 75th and 100th performance percentiles in the collected hyperparameter landscapes for the DQN subset.

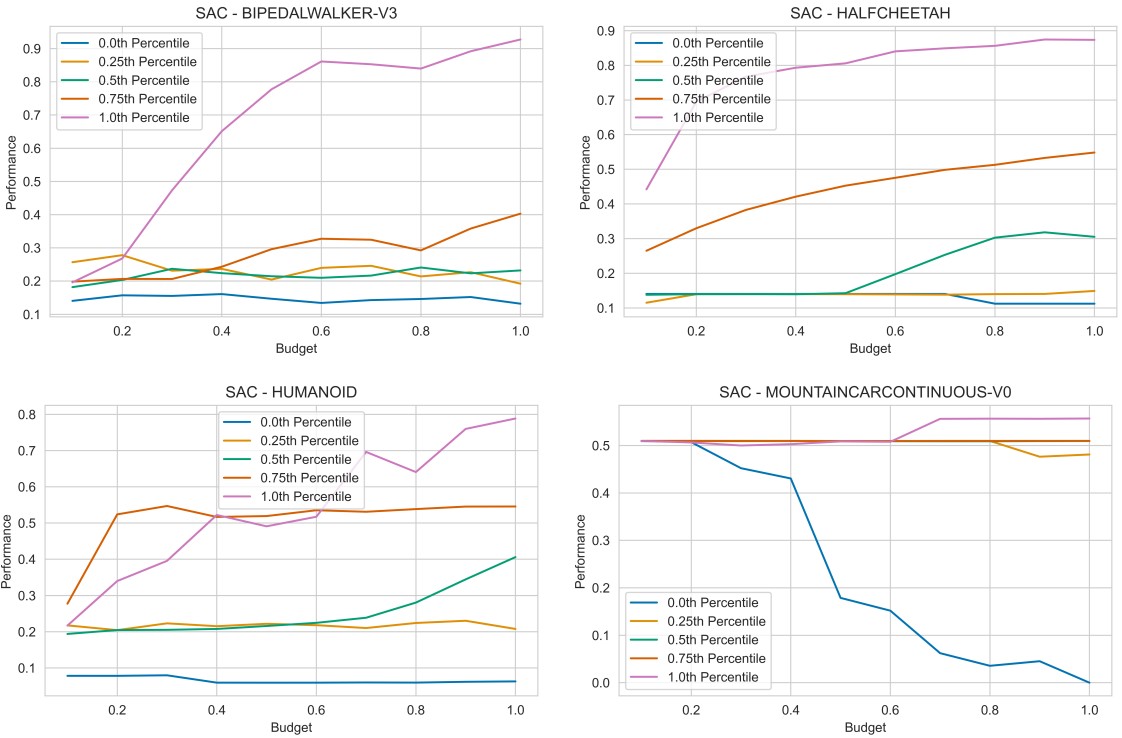

Figure 22: Learning curves of configurations at the 0th, 25th, 50th, 75th and 100th perfor-
mance percentiles in the collected hyperparameter landscapes for the SAC subset.

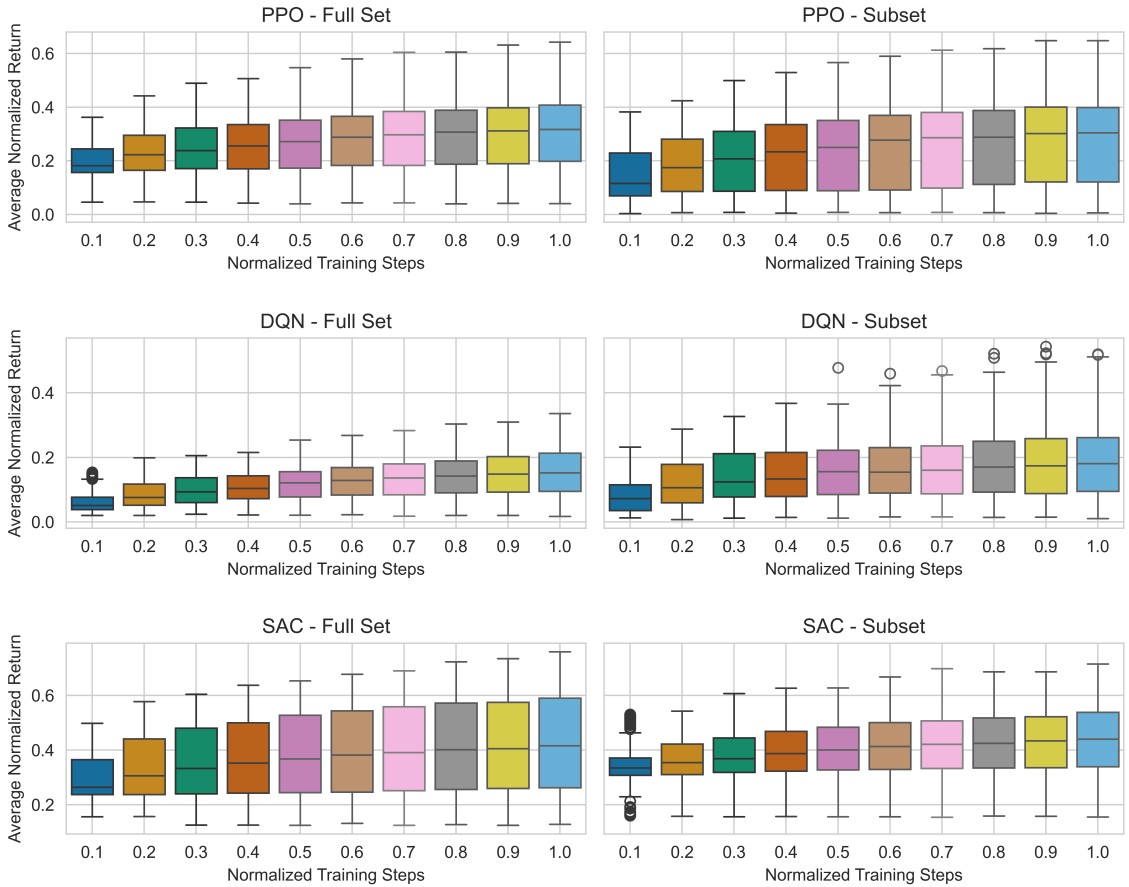

Figure 23: Performance distribution in the collected hyperparameter landscapes across all runs in the subset and full set at every 10% of training steps for PPO, DQN, and SAC.

## Appendix H. Hyperparameter Landscape Analysis

We used DeepCave (Sass et al., 2022) to analyze our performance dataset with regard to performance distribution and hyperparameter importance over time. Please note that in some cases, results can be missing, due to consistent numerical errors in the analysis, e.g., in the case of SAC on Halfcheetah.

### H.1 Landscape Behaviour

Algorithm configuration landscapes are often found to show relatively benign structure, characterized by unimodal responses and compensatory or negligible interactions (Pushak and Hoos, 2018). However, in our experiments, we observe that some partial ARLBench hyperparameter landscapes deviate from these traits, displaying challenging structure instead.

Figure 24 highlights the contrast between benign and adverse landscapes in our experiments, providing further insight into their differing characteristics.

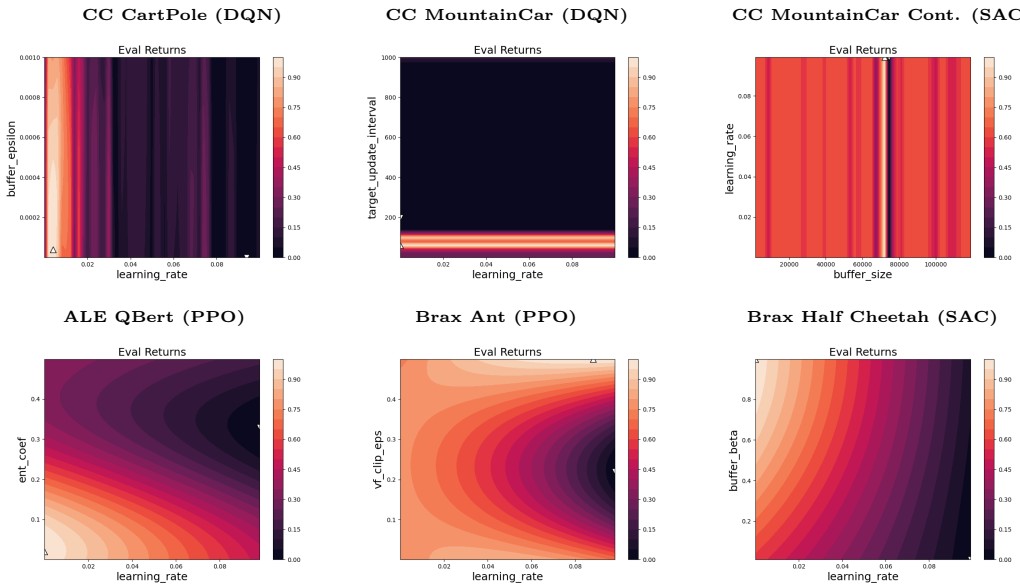

Figure 24: Comparison of adverse landscapes on the Top with typical benign landscapes on the Bottom. Lighter is better, mean performance over 10 seeds. Adverse landscapes exhibit multi-modality, whereas benign landscapes are uni-modal and display minimal to no hyperparameter interaction.

## H.2 Performance Distributions

Completing the results from the performance distributions comparison in Section 4.3, Figures 25 and 26 show the distribution of scores for the domains and subsets of DQN and SAC, respectively. Just like for PPO, there are fairly direct correspondences between selected environments and the score distributions of the full domains. The only seeming exception is Box2D for DQN, which has a lot of low scores that are not directly represented by one selected environment. Acrobot in that subset, however, covers a lot of such bad configurations even though it has higher performances overall.

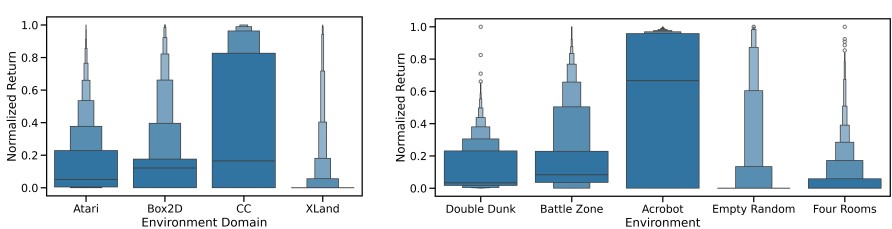

Figure 25: Return distributions across environment domains and the selected subset of DQN.

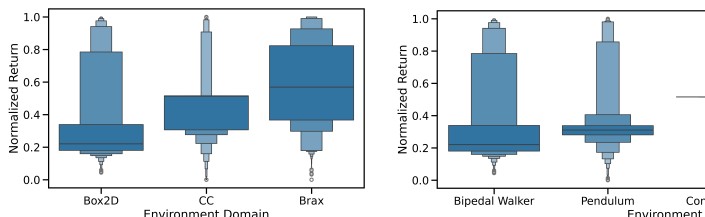

Figure 26: Return distributions across environment domains and the selected subset of SAC.

## H.3 Hyperparameter Importances

Tables 13, 14 and 15 show extended information on the number of important hyperparameters for each environment domain as well as the subset and full environment set.

| | ALE | Box2D | CC | XLand-Minigrid | Brax | All | Subset |
|---|---|---|---|---|---|---|---|
| #HPs with over 10% importance | 1.6 | 1.5 | 1.0 | 1.75 | 0.75 | 1.3 | 1.0 |
| #HPs with over 5% importance | 1.6 | 1.5 | 3.4 | 2.25 | 1.75 | 2.2 | 1.2 |
| #HPs with over 3% importance | 2.0 | 3.0 | 4.0 | 2.5 | 3.0 | 2.9 | 2.0 |

Table 13: Fraction of hyperparameters with importances on the full set and subset for PPO.

| | ALE | Box2D | CC | XLand-Minigrid | All | Subset |
|---|---|---|---|---|---|---|
| #HPs with over 10% importance | 2.0 | 1.0 | 1.0 | 2.25 | 1.77 | 2.0 |
| #HPs with over 5% importance | 2.8 | 1.0 | 1.33 | 3.5 | 2.54 | 2.75 |
| #HPs with over 3% importance | 3.8 | 2.0 | 2.33 | 4.0 | 3.38 | 3.5 |

Table 14: Fraction of hyperparameter importances on the full set and subset for DQN.

| | Box2D | CC | Brax | All | Subset |
|---|---|---|---|---|---|
| #HPs with over 10% importance | 1.0 | 1.5 | 1.0 | 1.14 | 0.75 |
| #HPs with over 5% importance | 1.0 | 3.0 | 1.5 | 1.86 | 1.25 |
| #HPs with over 3% importance | 1.0 | 3.5 | 2.75 | 2.71 | 2.5 |

Table 15: Fraction of hyperparameter importances on the full set and subset for SAC.

## Appendix I. Resource Consumption

All running time results are stated in Table 16 and were obtained using the same setup on the H100 cluster as described in Appendix B.1.

| Algorithm | Environment | Platform | Running Time [s] | Total Running Time [h] |
|---|---|---|---|---|
| DQN | Acrobot-v1 | CPU | 26.1 | 37.13 |
| DQN | BattleZone-v5 | GPU | 2967.69 | 4220.72 |
| DQN | CartPole-v1 | CPU | 10.27 | 14.6 |
| DQN | DoubleDunk-v5 | GPU | 2918.08 | 4150.16 |
| DQN | LunarLander-v2 | CPU | 34.47 | 49.03 |
| DQN | MiniGrid-DoorKey-5x5 | CPU | 81.44 | 115.82 |
| DQN | MiniGrid-EmptyRandom-5x5 | CPU | 30.32 | 43.12 |
| DQN | MiniGrid-FourRooms | CPU | 172.31 | 245.07 |
| DQN | MiniGrid-Unlock | CPU | 94.68 | 134.65 |
| DQN | MountainCar-v0 | CPU | 19.4 | 27.59 |
| DQN | NameThisGame-v5 | GPU | 2970.15 | 4224.22 |
| DQN | Phoenix-v5 | GPU | 2710.29 | 3854.64 |
| DQN | Qbert-v5 | CPU | 2943.79 | 4186.73 |
| PPO | Acrobot-v1 | CPU | 15.34 | 21.82 |
| PPO | BattleZone-v5 | GPU | 1154.29 | 1641.66 |
| PPO | BipedalWalker-v3 | CPU | 89.83 | 127.76 |
| PPO | CartPole-v1 | CPU | 7.95 | 11.3 |
| PPO | DoubleDunk-v5 | GPU | 1083.08 | 1540.38 |
| PPO | LunarLander-v2 | CPU | 162.97 | 231.78 |
| PPO | LunarLanderContinuous-v2 | CPU | 300.47 | 427.33 |
| PPO | MiniGrid-DoorKey-5x5 | CPU | 81.23 | 115.52 |
| PPO | MiniGrid-EmptyRandom-5x5 | CPU | 26.37 | 37.5 |
| PPO | MiniGrid-FourRooms | CPU | 179.84 | 255.77 |
| PPO | MiniGrid-Unlock | CPU | 112.33 | 159.76 |
| PPO | MountainCar-v0 | CPU | 13.21 | 18.79 |
| PPO | MountainCarContinuous-v0 | CPU | 7.68 | 10.93 |
| PPO | NameThisGame-v5 | GPU | 1130.46 | 1607.76 |
| PPO | Pendulum-v1 | CPU | 13.81 | 19.64 |
| PPO | Phoenix-v5 | GPU | 955.17 | 1358.46 |
| PPO | Qbert-v5 | CPU | 1145.07 | 1628.54 |
| PPO | ant | GPU | 220.87 | 314.13 |
| PPO | halfcheetah | GPU | 851.99 | 1211.73 |
| PPO | hopper | GPU | 458.43 | 651.98 |
| PPO | humanoid | GPU | 338.6 | 481.57 |
| SAC | BipedalWalker-v3 | CPU | 486.32 | 691.66 |
| SAC | LunarLanderContinuous-v2 | CPU | 381.22 | 542.17 |
| SAC | MountainCarContinuous-v0 | CPU | 557.13 | 792.36 |
| SAC | Pendulum-v1 | CPU | 111.76 | 158.95 |
| SAC | ant | GPU | 824.95 | 1173.26 |
| SAC | halfcheetah | GPU | 2194.59 | 3121.2 |
| SAC | hopper | GPU | 1263.28 | 1796.66 |
| SAC | humanoid | GPU | 871.83 | 1239.93 |

Table 16: Running times of algorithms and environments and respective platforms they were executed on. Column *Running Time* represents the duration of a single training session. *Total Running Time* indicates the cumulative hours spent on all experiments conducted. Each experiment was run for 4096 total runs, resulting in a total CPU running time of 10 105.34 h and GPU running time of 32 588.46 h (including 40.54 GPU hours for measuring the running times).

