# OpenReview forum: "ARLBench: Flexible and Efficient Benchmarking for Hyperparameter Optimization in Reinforcement Learning"
_DMLR — Accepted by DMLR_

### Review · Reviewer_XpS7 · 2025-06-05

**Recommendation:** 3
**Confidence:** 2

**Summary Of Contributions:**

This paper introduces ARLBench, a novel benchmark designed to support research in hyperparameter optimization (HPO) for reinforcement learning (RL). The authors identify inefficiencies and inconsistencies in existing HPO evaluation practices in RL and address them by developing a scalable, configurable, and dynamic benchmarking suite. ARLBench supports three widely-used RL algorithms (DQN, PPO, SAC) with JAX-based reimplementations and covers a wide range of environments. By selecting representative subsets of environments and configurations, the benchmark drastically reduces the computational cost of HPO evaluation while maintaining fidelity to broader performance trends. The paper also includes a publicly released large-scale meta-dataset and an interface supporting static and dynamic HPO methods.

**Strengths:**

See **Strengths And Weaknesses**.

**Audience:**

Yes

**Broader Impact Concerns:**

The paper does include a Broader Impact Statement that addresses its ethical implications.

**Claims And Evidence:**

This paper's claims are strongly substantiated by comprehensive empirical evidence. For example, ARLBench supports a wide range of HPO paradigms including static and dynamic methods.

**Datasets And Benchmarks:**

Yes

**Extended Submissions:**

NA

**Limitations:**

**Limited Algorithm Coverage**

The benchmark currently supports only three model-free RL algorithms: DQN, PPO, and SAC. While these are widely used, they do not represent the full spectrum of RL methods, such as model-based RL, offline RL, or multi-agent RL. Extending support to these areas would improve the generalizability and utility of ARLBench.

**Requested Changes:**

It is recommended to include additional RL algorithms to further strengthen the core contribution of this work.

**Strengths And Weaknesses:**

**Strengths**

- Timely and Relevant Contribution: HPO in RL is under-explored compared to supervised learning; this work directly addresses the benchmarking gap with practical tools and rigorous analysis.

- Efficiency and Accessibility: The benchmark achieves significant computational savings (up to 11.6× speedup) over prior baselines (e.g., SB3), enabling broader participation in AutoRL research.

- Open-Sourced and Well-Maintained: The project is publicly available with clear documentation, and the authors provide a future maintenance plan and community engagement path.

**Weaknesses**

- Scope Limitation: The benchmark currently supports only three model-free RL algorithms; extending to model-based RL, multi-agent RL, or offline RL remains an open task.

---

### Review · Reviewer_wjj1 · 2025-06-06

**Recommendation:** 3
**Confidence:** 2

**Summary Of Contributions:**

ARLBench introduces a flexible, efficient benchmark for hyperparameter optimization in reinforcement learning. It supports large configuration spaces, dynamic hyperparameter schedules, and broad RL environment coverage. By selecting representative environment subsets, it enables reproducible and low-cost evaluations, advancing AutoRL research through a publicly available dataset and framework tailored for scalability and extensibility.

**Strengths:**

The submission presents a significant and timely contribution to the AutoRL community by introducing ARLBench, a flexible and efficient benchmark for hyperparameter optimization in RL. It enables reproducible, low-cost evaluation across diverse algorithms and environments, addressing a major gap in existing HPO benchmarks. The methodology is sound, clearly described, and well-motivated by extensive empirical evaluation. The benchmark's design supports large configuration spaces and dynamic tuning strategies, aligning well with real-world needs.

**Audience:**

Yes

**Claims And Evidence:**

Yes

**Datasets And Benchmarks:**

N/A

**Extended Submissions:**

No

**Limitations:**

ARLBench is limited to model-free RL algorithms and does not yet support advanced algorithmic variants or model-based approaches.

**Requested Changes:**

Expanding to include/discuss model-based methods and broader task diversity would strengthen applicability.

**Strengths And Weaknesses:**

ARLBench is a timely and well-executed contribution, offering a flexible, efficient benchmark for HPO in RL with strong support for dynamic configurations and diverse tasks. However, its scope is limited to model-free algorithms, and the reliance on fixed environment subsets may restrict its applicability to broader or evolving RL scenarios.

---

### Review · Reviewer_cJYy · 2025-10-14

**Recommendation:** 4
**Confidence:** 2

**Summary Of Contributions:**

ARLBench introduces a large-scale, open benchmark for hyperparameter optimization (HPO) in reinforcement learning (RL).

It provides:

- Executable JAX-based implementations of three major model-based families of algorithms (DQN, SAC, PPO).
- A public meta-dataset containing thousands of training runs across domains (Atari, Brax, XLand, etc).

The work aims to turn RL HPO into a standardized, reproducible, and computationally tractable benchmark for evaluating both static and dynamic AutoRL methods.

**Strengths:**

See Strengths and Weaknesses section.

**Audience:**

Yes

**Broader Impact Concerns:**

No ethical or societal concerns are apparent.
The benchmark promotes reproducibility and openness in RL research.
Potential compute usage is substantial for full sweeps, but mitigated by the subset-selection efficiency gains.
No privacy, fairness, or data-collection risks exist since all environments are synthetic.

**Claims And Evidence:**

All major claims are empirically supported:

- Rank correlation between subset and full pool is shown with cross-validation.
- Landscape structure preservation is demonstrated via fANOVA consistency.

Some secondary claims (temporal fidelity, generalization) are suggestive but not fully validated, warranting cautious interpretation

**Datasets And Benchmarks:**

The dataset component is well-documented and responsibly released, with clear meta-data, versioning and maintenance described in Appendix C.

**Extended Submissions:**

This is not an extended version of prior work.

**Limitations:**

- ARLBench’s subset representativeness is validated only for the included algorithms and hyperparameter ranges; transfer to novel RL methods is untested
- It assumes final-return rank fidelity as the measure of benchmark equivalence, the authors do not examine temporal learning-curve similarity

**Requested Changes:**

- Add temporal-fidelity evidence: including a plot or statistic showing correlation of early-budget vs. late-budget performance between subset and full set could be helpful
- Outline plans for adding model-based, offline, and/or multi-agent tracks

**Strengths And Weaknesses:**

**Strengths**:
- To the best of my knowledge, ARLBench is the first standardized, publicly released, executable benchmark for hyperparameter optimization in RL that supports variable budgets and checkpoint/resume functionality, extending prior static efforts (e.g., HPO-RL-Bench)
- Empirical validation is well designed
- JAX implemenations yield significant speedups
- Highly relevant; directly addresses a key bottleneck for AutoRL and dynamic HPO research

**Weaknesses**:
- Only covers model-free, single-agent, online RL
- Optimized for existing algorithms; may not generalize to novel RL methods or new hyperparameter semantics
- *(Important)*: Subsets preserve final ranking but not learning-curve similarity or early-stopping behaviour. This limits ARLBenchmark's viability for budgeted HPO algorithms:
  - Since many methods allocate budgets adaptively and stop poorly performing runs early based on partial progress, if early performance rankings differ between subset and full set, resource allocation decisions will diverge